# LIGHTRETRIEVER: A LLM-BASED TEXT RETRIEVAL ARCHITECTURE WITH EXTREMELY FASTER QUERY INFERENCE

**Guangyuan Ma**[1,2]**, Yongliang Ma**[3]**, Xuanrui Gou**[2]**, Zhenpeng Su**[1,2]**, Ming Zhou**[3]**, Songlin Hu**[1,2*]

[1]Institute of Information Engineering, Chinese Academy of Sciences, Beijing, China
[2]School of Cyber Security, University of Chinese Academy of Sciences, Beijing, China
[3]Langboat Technology, Beijing, China
{maguangyuan,suzhenpeng,husonglin}@iie.ac.cn, gouxuanrui21@mails.ucas.ac.cn,
{mayongliang,zhouming}@langboat.com

## ABSTRACT

Large Language Models (LLMs)-based text retrieval retrieves documents relevant to search queries based on vector similarities. Documents are pre-encoded offline, while queries arrive in real-time, necessitating an efficient online query encoder. Although LLMs significantly enhance retrieval capabilities, serving deeply parameterized LLMs slows down query inference throughput and increases demands for online deployment resources. In this paper, we propose **LightRetriever**, a novel LLM-based retriever with extremely *lightweight* query encoders. Our method retains a full-sized LLM for document encoding, but reduces the workload of query encoding to no more than an embedding lookup. Compared to serving a full LLM on an A800 GPU, our method achieves over 1000x speedup in query encoding and over 10× increase in end-to-end retrieval throughput. Extensive experiments on large-scale retrieval benchmarks show that LightRetriever generalizes well across diverse tasks, maintaining an average of 95% retrieval performance.

## 1 INTRODUCTION

Given search queries and candidate documents, text retrieval employs dual-encoder architectures to encode representation vectors and search documents relevant to queries based on query-document similarities (Salton et al., 1975). The encoded vectors could be condensed into low-dimensional vectors for dense retrieval or sparsified into high-dimensional vectors for sparse retrieval. Dense retrieval (Karpukhin et al., 2020) compresses textual information into dense vectors with parameterized language models (LMs) and retrieves based on inner product similarity. Meanwhile, sparse retrieval (Robertson et al., 1994) characterizes sparse term frequencies on a much broader vocabulary space, and then searches based on lexical overlap and term-based relevance metrics, such as direct TF-based search (Inner product) (Bai et al., 2020; Formal et al., 2021) and BM25 (Robertson et al., 1994). Additionally, hybrid retrieval enhances retrieval abilities by interpolating their similarity scores (Wang et al., 2021) or ranks (Cormack et al., 2009). Text retrieval is broadly applied in various scenarios, such as web search (Lu et al., 2022), semantic textual similarity (Bowman et al., 2015), fact checking (Thorne et al., 2018), and retrieval augmented generation (Lewis et al., 2020).

Deploying retrieval systems requires different processing workflows for queries and documents. As shown in Figure 1, as search candidates, documents are encoded with a document encoder $Enc_d$ and then indexed into search databases (Johnson et al., 2021; Robertson et al., 1994). Because documents are obtained before the search operation, they are pre-computable and distributable, which can be indexed offline and distributed anywhere (Yang et al., 2018a). On the contrary, queries are input and encoded in real-time, which are non-pre-computable and non-distributable. The query encoder $Enc_q$ needs to be served online. Special GPU accelerators are mandatory if the query encoder is a deeply parameterized LM. Recent state-of-the-art (SOTA) retrievers (Wang et al., 2024b;

---

*Corresponding author. Code is available at https://github.com/caskcsg/lightretriever.

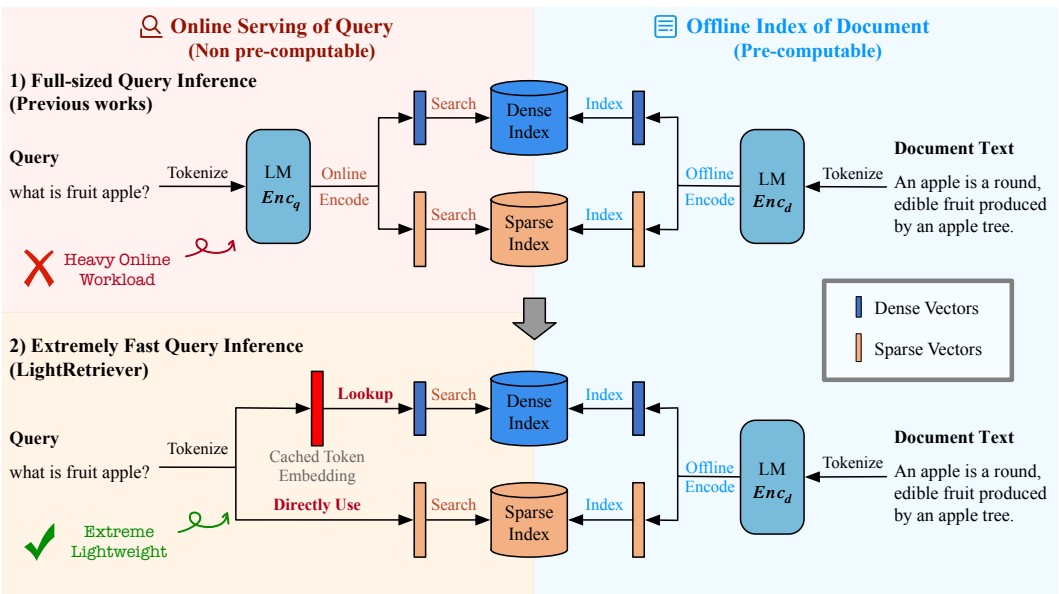

Figure 1: LightRetriever targets at extreme query inference speed for LLM-based text retrieval, reducing the workload of query encoding to no more than an embedding lookup.

BehnamGhader et al., 2024) employ deeply parameterized LLMs as symmetric dual-encoders $Enc_q$ and $Enc_d$ for enhanced retrieval capabilities. However, serving giant LLMs slows down query inference throughput and depletes online deployment resources (Kwon et al., 2023). Despite the rapid progress on improving retrieval quality, little prior work has investigated the efficiency challenges of LLM-based retrievers, necessitating special optimizations for query inference efficiency.

Modern Transformer-based LMs empower text retrievers with rich contextual modeling through the attention mechanism (Vaswani et al., 2017), enabling deep interactions across contextual tokens. This capability explains the success of recent LLM-based dual-encoders (Wang et al., 2024b; BehnamGhader et al., 2024), but it also means that both queries and documents incur the same computational burden. In contrast, traditional statistical methods such as BM25 (Robertson et al., 1994) rely purely on lexical overlap and token frequency, requiring negligible inference cost while still remaining competitive in many retrieval tasks (Thakur et al., 2021). This contrast highlights a practical tension: While document encoders can afford heavy pre-computation offline, query encoders must operate under strict online latency and resource constraints. Therefore, it is natural to ask: do queries truly need the same degree of deep contextual modeling as documents?

To strike a balance between performances and efficiencies, we propose **LightRetriever**, a new hybrid retrieval architecture that explicitly breaks the symmetry between document and query encoders. Our key insight is that while documents benefit from the full modeling power of LLMs, queries do not require equally heavy processing to achieve strong retrieval performance. To this end, LightRetriever preserves a full-sized LLM encoder on the document side, but *entirely removes the deep modeling on the query side*. 1) For dense retrieval, we end-to-end train cacheable token embeddings from the LLM-based query encoder and aggregate them through simple averaging, reducing query inference to a single embedding lookup; 2) For sparse retrieval, we directly map token counts to sparse vectors without any encoding. Dense and sparse scores are linearly interpolated (Wang et al., 2021) for the final hybrid retrieval.

LightRetriever achieves retrieval quality mostly comparable to full dual-encoder LLMs, but with orders-of-magnitude faster query inference. Compared to serving a full-sized LLM on an A800 GPU, our method achieves over a thousand times speedup for query encoding, and over 10x total throughput improvements. Importantly, our experiments show that adopting a lightweight query encoder does not mean queries are deprived of deep semantic understanding. Our work tries to transfer the major modeling cost from the query side to the document side, while still preserving the benefits

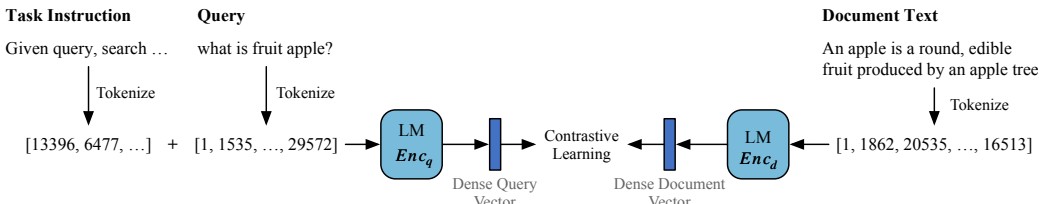

Figure 2: Contrastive training of full-sized symmetric dense retrieval. A full-sized dual-encoder is used to model both queries and documents. Task-specific instructions are added as common practices (Su et al., 2023; Wang et al., 2024b) to promote better domain adaptation abilities.

of LLM-powered relevance-based similarity search with query-document interactions. Experiments on large-scale retrieval benchmarks demonstrate that our method generalizes well across diverse LLMs (including Llama-1B, 3B, 8B, and Qwen-1.5B, 3B, 7B) and retrieval benchmarks (including 15 English tasks from BeIR (Thakur et al., 2021) and 8 Chinese tasks from CMTEB Retrieval (Xiao et al., 2024)), maintaining an average of 95% retrieval performance.

## 2 ALGORITHM

### 2.1 LM-BASED TEXT RETRIEVAL

**Definition** Given a search query $q$ and candidate documents $d = \{d^+; d^-\} \in \mathcal{N}(q)$ (including irrelevant documents $d^-$ and relevant documents $d^+$) retrieved from indexes $\mathcal{N}$, LM-based text retrieval aims to learn optimized encoders $Enc_q$ and $Enc_d$ with parameters $\theta$, where the similarity of the query vector $v_q = Enc_q(q)$ and irrelevant document vectors $v_{d^-} = Enc_d(d^-)$ are encouraged to be smaller than those of relevant documents $v_{d^+} = Enc_d(d^+)$. Assuming the dot product is used as the similarity metric, the above optimization goals can be described as the following objective,

$$\min_{\theta} \max_{d^- \in \mathcal{N}(q)} \mathcal{E}(q, d^+, d^-; \theta), \text{ where } \mathcal{E}(q, d^+, d^-; \theta) = v_q \cdot v_{d^-} - v_q \cdot v_{d^+} \tag{1}$$

which lowers the upper boundary of ranking similarity errors $\mathcal{E}$ (Chen et al., 2009). The above objective can be effectively optimized by contrastive loss (Cao et al., 2007), margin loss (Rosset et al., 2003), etc. Following (Karpukhin et al., 2020), listwise contrastive loss ($\ell^{CL}$) is used in our work,

$$\ell^{CL} = -\log \frac{e^{v_q \cdot v_{d^+}/\tau}}{e^{v_q \cdot v_{d^+}/\tau} + \sum_{d^- \in \mathcal{N}(q)} e^{v_q \cdot v_{d^-}/\tau}}. \tag{2}$$

, where $\tau$ is the temperature. The retrieval capacity is strongly correlated with parametrized encoders. As discussed before, serving deep parametrized LMs, especially giant LLMs, consumes large amounts of deployment resources. Thus, our work aims to address this issue by designing a novel asymmetric retrieval architecture with extreme imbalance.

**Instructed retrieval** Similar to instructed fine-tuning (SFT) (Ouyang et al., 2022), modern LM-based retrievers (Su et al., 2023; Xiao et al., 2024; Wang et al., 2024a), especially LLM retrievers (Wang et al., 2024b; BehnamGhader et al., 2024; Lee et al., 2024), customize task specific instructions for better domain adaption abilities. Specifically, as shown in Figure 2, the task instruction is prepended to the queries before retrieval. Although (Su et al., 2023; Xiao et al., 2024) use instructions on both query and passage sides, most LLM-based retrievers (Wang et al., 2024b; BehnamGhader et al., 2024; Lee et al., 2024) do not use passage instructions to reuse the pre-built indexes across different tasks. In our work, dense retrieval follows the query instructions of Mistral-E5 (Wang et al., 2024b) and does not involve passage instructions. The sparse retrieval is incompatible

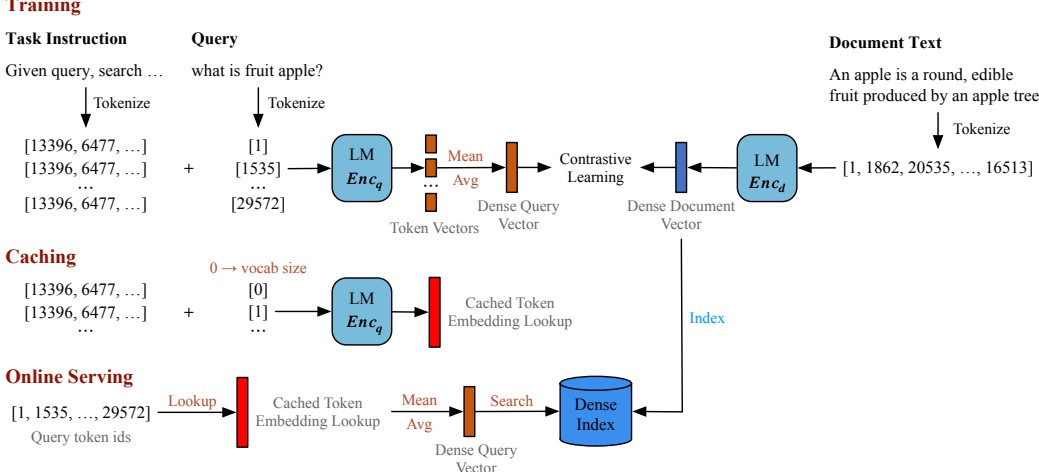

Figure 3: Dense retrieval of LightRetriever. Three stages are involved for efficient query modeling. 1) **Training**: A shared prompt concatenated with a single query token is passed through a *full encoder* independently to obtain token-level representations. The full query representation is then computed by averaging the token vectors corresponding to each query token. 2) **Caching**: Prior to serving, the trained encoder is used to precompute token embeddings for the entire LLM's vocabulary, which are stored in a single embedding lookup matrix. 3) **Online Serving**: At inference time, query embeddings are efficiently generated by lookup and averaging the cached token embeddings, eliminating the need for deep model inference.

with instructions because it does not involve learnable query encoders. Detailed instructions are shown in the Appendix A.7.

## 2.2 DENSE RETRIEVAL OF LIGHTRETRIEVER

For LLM-based retrieval, LightRetriever pushes the query inference efficiency to a new extreme by completely removing the deep modeling on the query side. For dense retrieval, our work end-to-end trains cacheable token vectors for online serving.

**Training**   Given a task instruction $Inst$ and query tokens $T_q = \{t_0, t_1, ..., t_{n-1}\}$, each query token $t_i$ ($i \in \{0, ..., n-1\}$) are prepended with the instruction. Then, the corresponding query token vectors $v_{t_i}^{\text{den}}$ are obtained via last token pooling with the query encoder[1]. Then, the dense query vector $v_q^{\text{den}}$ of the whole query text is aggregated by averaging token vectors,

$$v_q^{\text{den}} = \frac{1}{n} \sum_{i=0}^{n-1} v_{t_i}^{\text{den}}, \text{ where } v_{t_i}^{\text{den}} = Enc_q(Inst; t_i) \tag{3}$$

Contrastive loss is the main objective for dense training, as defined in Equation 2.

**Caching**   Query tokens do not deeply interact with each other. Thus, the query token vectors are cacheable. Given a certain vocabulary with size $V$, all token vectors $v_t^{den}, t \in \{0, ..., V-1\}$ are precomputed as in Equation 3 and stored in a lookup Embedding $E$. Typically, caching such a lookup Embedding on 8 H800 with a Llama-8b model consumes less than 20s, which is a one-go and cheap operation. In Appendix A.4, LightRetriever also shows good ability to balance the serving Embedding size via simple dimension truncations (Kusupati et al., 2022).

---

[1] A customized causal mask is designed to avoid repeated prompt computations, detailed in Appendix A.8.

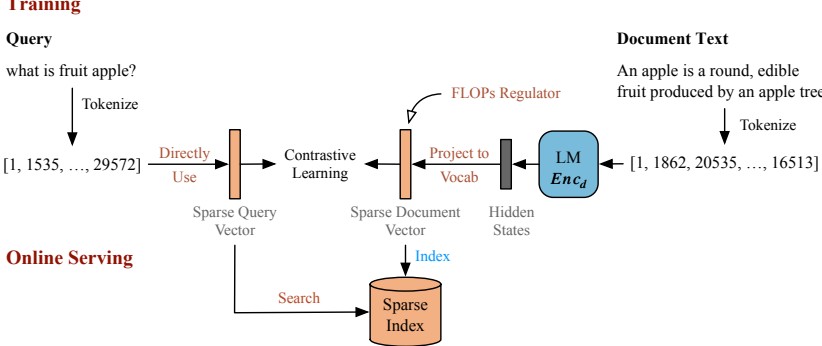

Figure 4: Sparse retrieval of LightRetriever. The query representation is a term-based, unlearnable sparse vector, which is obtained from tokenization. The document representation is end-to-end trained by projecting last layer hidden states of LLM to vocabulary space, then optimized via contrastive learning.

**Online serving** LLM-based encoders are no longer needed during online serving. Instead, only one Embedding layer, with a single weight matrix shaped $[V, H]$, is hosted in RAM. $H$ is the dimension of LLM's hidden states, which is also the dimension of dense vectors. The dense query vector is obtained with a simple lookup-then-average operation, which is super efficient with or without GPU acceleration.

$$v_q^{\text{den}} = \frac{1}{n} \sum_{i=0}^{n-1} v_{t_i}^{den} = \frac{1}{n} \sum_{i=0}^{n-1} E[t_i] \tag{4}$$

## 2.3 SPARSE RETRIEVAL OF LIGHTRETRIEVER

The sparse retrieval of LightRetriever steps towards lightweight even further, by completely removing the need for the sparse query encoder.

**Training** The sparse query vector $v_q^{spr}$ directly maps a token and its corresponding quantity,

$$v_q^{\text{spr}}[t] = \begin{cases} 0 & \text{if } t \notin T_q \\ \text{Number of } t & \text{if } t \in T_q \end{cases} \tag{5}$$

where $t \in \{0, ..., V - 1\}$ represents valid token ids within LLM's vocabulary. We obtain the sparse document vector $v_d^{\text{spr}}$ by first feeding the document $d$ into the pretrained language model (LM), resulting in the final-layer hidden states $h_{\text{last}} = LM(d)$. These hidden states are then projected into the vocabulary space to get logits $w$ via the model's own language modeling head, parameterized by a projection matrix $P$. To induce sparsity and suppress dominant term frequencies, we apply a ReLU activation followed by a log-saturation function, and then perform a max-pooling operation of the logits $w$ over the sequence length dimension by following (Formal et al., 2021),

$$v_d^{\text{spr}} = \max(w), \text{ where } w = \text{logits} = \ln(\max(h_{\text{last}} \cdot P, 0) + 1) \tag{6}$$

Here, $v_d^{\text{spr}} \in \mathbb{R}^V$ denotes the final sparse representation vector of the document. Given sparse vectors $v_q^{\text{spr}}$ and $v_d^{\text{spr}}$, the same contrastive loss in Equation 2 is applied for representation learning.

$v_d^{\text{spr}}$ is not naturally sparse enough, because LM tends to generate dense distributions. Thus, the sparsify regulator is needed during the training phase. Given a batch of documents $d = \{d_0, ..., d_i\}, i < $ batch size $N$, the FLOPs regulator (Paria et al., 2020) is used for sparsification. This regulator first averages the logits across different documents within a certain batch, then computes the squared sum to get the final regulator.

$$\ell_{\text{FLOPS}} = \sum_{t=0}^{V-1} \left( \frac{1}{N} \sum_{i=0}^{N-1} w_t^{(d_i)} \right)^2 \tag{7}$$

**Online serving**  The sparse query vector $v_q^{\text{spr}}$ during online serving follows the same rule in Equation 5. No more query encoder for sparse retrieval is needed.

## 3 EXPERIMENTS

### 3.1 SETTINGS

**Training data**  Our work finetunes on a large existing collection of 20 English and 3 Chinese datasets with 8.38M samples to ensure broad domain coverage and diversity. To ensure reproducibility, our work reuses the preprocessed fine-tuning data from tDRO (Ma et al., 2025) and Sentence Transformers Training Data. English datasets include Amazon Review (Ni et al., 2019), Eli5 (Fan et al., 2019), FEVER (Thorne et al., 2018), FiQA (Maia et al., 2018), GooAQ (Khashabi et al., 2021), HotpotQA (Yang et al., 2018b), MSMARCO (Nguyen et al., 2016), NFCorpus (Boteva et al., 2016), NPR (Lucas et al., 2023), NQ (Kwiatkowski et al., 2019), PAQ (Lewis et al., 2021), Quora Duplicates (Iyer et al., 2017), S2ORC (Lo et al., 2020), Scifact (Wadden et al., 2020), Specter (Cohan et al., 2020), StackExchange Duplicates (da Silva et al., 2018), E5 synthetic (BeastyZ, 2024), Trivia (Joshi et al., 2017), WikiHow (Koupaee & Wang, 2018), and Yahoo Answers (Zhang et al., 2015). Chinese datasets include cMedQA2 (Zhang et al., 2018), DuReader (Qiu et al., 2022), and T2Ranking (Xie et al., 2023).

**Benchmarks**  BeIR (Thakur et al., 2021) and CMTEB Retrieval (Xiao et al., 2024) benchmarks are used in the evaluation. BeIR is a massive English retrieval benchmark collection with 15 evaluation sets. These heterogeneous datasets cover different domains, tasks, and retrieval granularities. CMTEB Retrieval is a massive Chinese retrieval benchmark collection with 8 sets, also covering a variety of domains. nDCG@10 is reported as the main metric by following (Thakur et al., 2021; Xiao et al., 2024), accessing both recall and ranking abilities. Recalls (R@{20, 50, 100}) are also reported to assess retrieval ability at large windows.

**Training hyper-parameters**  Our experiments are conducted based on the implementation of tDRO (Ma et al., 2025). All experiments are performed with a batch size of 128, 7 hard negatives, a max sequence length of 512, a contrastive temperature $\tau$ of 0.02, and 12k total steps. Following SPLADE (Formal et al., 2021), the coefficient of the FLOPs regulator is 0.001, and the regulator is quadratically increased to the maximum at the first 4k steps, which helps reduce the side effects of the regulator at initial steps (Paria et al., 2020). LoRA (Hu et al., 2022) is used to save training GPU memory, where LoRA r is 16, alpha is 32, and dropout is 0.1. Different LLMs with different sizes are tested as the backbone encoders to verify the generalization abilities, including Llama-3.2-1B, 3B, Llama-3.1-8B (Touvron et al., 2023), and Qwen-2.5-1.5B, 3B, 7B (Bai et al., 2023). The hybrid similarity scores are linearly summed from normalized dense and sparse scores (Wang et al., 2021). All trainings are conducted on 8 NVIDIA H800 or A800 GPUs. Specifically, training a LightRetriver-Llama8b on 8 H800 consumes about 15 hours.

**Speed comparisons**  To evaluate the online speedup of LightRetriever, we compare the online time consumptions for retrieving 65536 Bing queries over 1M passages from MSMARCO (Nguyen et al., 2016). All speed tests are conducted on a single A800 GPU, with a batch size of 256, a dimension of 1k for Faiss exact search, and 64 threads for Anserini (Lucene) sparse search. Faiss and Anserini search in parallel, where Faiss searches more slowly in our cases.

**Performance comparisons**  To comprehensively evaluate the performance cost of removing deep query modeling, we trained a series of fully symmetric retrievers under identical training conditions and compared their retrieval performance degradation. Additionally, we benchmarked several recent dense, sparse, and hybrid retrievers from prior work. For dense retrieval, we report results from Sentence Transformers Static Embedding (Aarsen, 2025), distilled USTAD (Kim et al., 2024), E5-Mistral (Wang et al., 2024b), and LLM2Vec (BehnamGhader et al., 2024). For sparse retrieval, we

Table 1: Comparison of retrieval speed and effectiveness for full symmetric retrievers, one Llama-8b layer, and the query-lightweight LightRetriever. For retrieval efficiency, we report the consumed time of: query tokenization, model encoding, maximum search time of Faiss/Lucene, and total end-to-end retrieval time. The overall throughput is measured in Queries Per Second (QPS). †Using the first Transformers layer of Llama-8b. *Significant speedup (p ≤ 0.01) over baselines.

| | Benchmark / nDCG@10 | | Time consumption / s & Throughput / QPS | | | | |
|---|---|---|---|---|---|---|---|
| Model | BeIR | CMTEB-R | Tokenize | Encode | Search | Total | QPS |
| Full-Llama8b | 56.8 | 67.6 | 1.3746 | 109.4853 | 8.5133 | 119.3730 | 549 |
| Full-Llama3b | 55.6 | 66.1 | 1.3250 | 52.5890 | 8.5040 | 62.4180 | 1050 |
| Full-Llama1b | 53.1 | 63.8 | 1.2564 | 19.9030 | 8.4897 | 29.6490 | 2210 |
| 1st-Layer of Llama8b† | 50.1 | 54.4 | 1.3393 | 2.3409 | 8.1634 | 11.8440 | 5533 |
| **LightRetriever-Llama8b** | 54.4 | 63.0 | 0.8209 | **0.0412**$^{*}$ | 8.5010 | **9.3630** | **6999**$^{*}$ |

include BM25 and English SPLADE-v3 (Lassance et al., 2024). For hybrid retrieval, we evaluate the interpolation of Static Embedding + BM25, and BGE-m3$_{dense+sparse}$ (Chen et al., 2024).

Notably, Static Embedding performs retrieval using a single shared embedding layer for both queries and documents, offering low retrieval performance but a highly efficient baseline. BM25, as a classical term-frequency-based method, remains a widely adopted sparse retriever. The combination of them thus provides a reference for efficient retrieval on both sides. In contrast, BGE-m3 is a Transformer-based retriever trained with multi-stage pretraining and fine-tuning, and represents state-of-the-art performance in hybrid retrieval. USTAD (Kim et al., 2024) reports a small 6-layer DistilBERT retriever, offering a SOTA distillation baseline for small language models (SLM). In Appendix A.5, applications of similar distillation techniques are also discussed in our works.

Except for Static Embedding and BM25, all compared retrievers, especially LLM retrievers, rely on deep Transformer encoders on both query and document sides. LightRetriever is, to our knowledge, the first method to address the online efficiency bottleneck on the query side of LLM-based retrievers, striking a balance between retrieval quality and inference efficiency.

## 3.2 MAIN RESULTS

**Encoding speeds**    Table 1 presents a detailed breakdown of retrieval time for processing 65,536 Bing queries over 1 million MSMARCO passages, highlighting the efficiency advantages of LightRetriever over full symmetric retrievers. Among all time consumptions, model encoding is the dominant contributor to latency in LLM-based retrievers. For example, Full-Llama8b and Full-Qwen7b require over 100 seconds solely for query encoding (109.5s and 100.7s, respectively), making them a bottleneck for large-scale, real-time retrieval.

In contrast, LightRetriever reduces encoding time to below 50 ms (0.0412–0.0420s), reaching a maximal 2500× speedup of full-sized Llama-8b in the encoding phase. This drastic reduction is made possible by replacing full forward passes of large Transformer models with a simple embedding lookup. And the overall throughput achieves over 10x QPS speedup. Notably, this lightweight encoding does not significantly compromise retrieval performance. LightRetriever maintains competitive nDCG@10 scores on both English (BeIR) and Chinese (CMTEB-Retrieval) benchmarks.

A potent baseline to improve LLM query encoder efficiency is layer reduction. We evaluated by training and serving only the first Transformer layer of the Llama3-8b model on the query side under the same settings, meanwhile retaining full-sized model on the document side. While this reduces model size during training, it underperforms the LightRetriever (–4.3 on BeIR, –8.6 on CMTEB-R) and introduces inference-time overhead of 2.34s. This proves the need for full-sized query modeling during the training phase.

The above results prove the effectiveness of our extreme decoupled modeling paradigm, which drastically improves online query encoding efficiency while preserving retrieval quality, making it suitable for latency-critical applications.

**Overall performance comparisons**    Table 2 compares retrieval effectiveness on English (BeIR) and Chinese (CMTEB-Retrieval) benchmarks. The main focus of this comparison is to evaluate the

Table 2: Performance comparisons on BeIR and CMTEB Retrieval (CMTEB-R) benchmarks. The best metrics of full symmetric retrievers and LightRetrievers are marked in bold. Their nDCG@10 gaps are also presented.

| Benchmark | BEIR (15 datasets) | | | | CMTEB-R (8 datasets) | | | |
|---|---|---|---|---|---|---|---|---|
| | nDCG@10 | R@20 | R@50 | R@100 | nDCG@10 | R@20 | R@50 | R@100 |
| Static Embed | 34.1 | 44.4 | 53.1 | 59.5 | 31.3 | 45.5 | 54.3 | 60.4 |
| BM25 | 41.7 | 48.8 | 56.5 | 61.8 | 50.8 | 63.9 | 70.0 | 74.1 |
| Static Embed + BM25 | 44.7 | 51.9 | 59.8 | 65.1 | 52.1 | 65.8 | 71.9 | 76.1 |
| USTAD | 44.2 | - | - | 65.3 | - | - | - | - |
| BGE-m3$_{dense+sparse}$ | 49.6 | 56.6 | 63.6 | 68.8 | 65.6 | 79.7 | 85.2 | 88.4 |
| SPLADE-v3 | 50.2 | 56.6 | 63.3 | 68.6 | - | - | - | - |
| LLM2Vec$_{llama8b}$ | 56.6 | 62.8 | 69.8 | 74.9 | 54.4 | 67.0 | 72.3 | 75.6 |
| E5-Mistral7b | **56.9** | 62.1 | 68.9 | 73.6 | 61.8 | 75.1 | 81.6 | 87.1 |
| **Full Symmetric Retrievers** | | | | | | | | |
| Llama3.2-1b | 53.1 | 60.2 | 66.9 | 72.0 | 63.8 | 78.1 | 84.1 | 87.9 |
| Llama3.2-3b | 55.6 | 62.9 | 69.7 | 74.7 | 66.1 | 81.0 | 86.6 | 90.0 |
| Llama3.1-8b | 56.8 | **64.3** | **71.1** | **75.7** | 67.6 | 82.0 | 87.4 | 90.9 |
| Qwen2.5-1.5b | 54.0 | 60.5 | 67.5 | 72.3 | 65.9 | 81.0 | 86.7 | 90.4 |
| Qwen2.5-3b | 54.9 | 62.0 | 68.7 | 73.5 | 69.4 | 83.8 | 88.9 | 91.8 |
| Qwen2.5-7b | 56.6 | 63.6 | 70.5 | 75.2 | **70.1** | **84.4** | **89.4** | **92.4** |
| **LightRetriever** | | | | | | | | |
| Llama3.2-1b | 52.0$_{-1.1}$ | 58.1 | 65.1 | 70.1 | 60.8$_{-3.0}$ | 74.8 | 81.2 | 85.0 |
| Llama3.2-3b | 53.5$_{-2.1}$ | 59.9 | 66.8 | 71.7 | 61.7$_{-4.4}$ | 76.3 | 82.3 | 86.3 |
| Llama3.1-8b | **54.4**$_{-2.4}$ | **60.9** | **67.7** | 72.8 | 63.0$_{-4.6}$ | 77.3 | 83.6 | 87.4 |
| Qwen2.5-1.5b | 52.1$_{-1.9}$ | 58.2 | 65.1 | 70.0 | 63.8$_{-2.1}$ | 78.3 | 84.5 | 88.3 |
| Qwen2.5-3b | 52.8$_{-2.1}$ | 59.1 | 66.0 | 70.9 | 65.7$_{-3.7}$ | 80.2 | 86.1 | 89.3 |
| Qwen2.5-7b | 53.8$_{-2.8}$ | 60.3 | 67.5 | 72.5 | **66.5**$_{-3.6}$ | **81.1** | **86.7** | **89.7** |

performance degradation introduced by LightRetriever's lightweight query encoding. Overall, by leveraging deep Transformer encoders on both query and document sides, full symmetric retrievers with deep LLM encoders achieve the best scores (e.g., Llama3.1-8b on BeIR with 56.8 nDCG@10, Qwen2.5-7b on CMTEB-R with 70.1 nDCG@10), but at a significant computational cost.

LightRetriever achieves promising effectiveness while drastically reducing query-side computational burden. Across different backbones, it incurs only modest degradations (typically around 5% with 1–5 absolute points). For example, LightRetriever-Qwen2.5-7b achieves 53.8 nDCG@10 on BeIR and 66.5 nDCG@10 on CMTEB-R, only 2.8 and 3.6 lower than its full counterpart.

Compared with prior LLM-based retrievers, LightRetriever is also competitive: LightRetriever-Llama3.1-8b (54.4 nDCG@10 on BeIR) surpasses BGE-m3$_{dense+sparse}$ (49.6 nDCG@10) and approaches LLM2Vec and E5-Mistral, despite avoiding full query inference. Moreover, case studies in Appendix A.3 show that LightRetriever is capable of understanding complex queries, such as compositional semantics and negation, rather than simple lexical matches. Overall, LightRetriever balances retrieval quality and efficiency, offering a scalable paradigm for latency-sensitive and high-throughput retrieval scenarios.

**Category-wise performance comparison** To provide a more fine-grained analysis of our query-lightweight proposal across different retrieval scenarios, we further categorize BeIR datasets into 8 task groups by following the original paper (Thakur et al., 2021) and summarize the results in Table 3. Overall, LightRetriever shows strong robustness on general relevance-based retrieval tasks, including Bio-Medical IR, Argument Retrieval, Fact Checking, Duplicate Question Retrieval, Passage Retrieval, and General QA, where it consistently preserves over 93% of the full symmetric retriever's effectiveness, and even surpasses it in some cases (e.g., TREC-COVID, ArguAna, and Touche2020). This indicates that lightweight query encoding is sufficient to capture the semantic signals in typical relevance-based ad-hoc and open-domain retrieval.

In contrast, moderately larger performance degradations are observed on more challenging tasks, such as Domain-specific QA (HotpotQA, FiQA), Entity Retrieval (DBPedia), and Citation Prediction (SCIDOCS), where the retention drops to around 87-89%. Additional benchmarks on CoIR (Li et al., 2025) and BRIGHT (Su et al., 2025) in §A.10 also show inferior performance of query-

Table 3: Category-wise performance comparison on BEIR. We report the average nDCG@10 of the full symmetric retriever (Llama-3.1-8B), LightRetriever (Llama-3.1-8B), and the performance retention of LightRetriever (LightRetriever/Full-Symmetric).

| Task | Datasets | Full | Light | Retention |
|------|----------|------|-------|-----------|
| Bio-Medical IR | TREC-COVID | 62.0 | 71.5 | 115.3% |
| Argument Retrieval | ArguAna, Touche2020 | 39.7 | 42.5 | 107.2% |
| Fact Checking | FEVER, ClimateFEVER, SciFact | 70.8 | 67.6 | 95.5% |
| Duplicate Question Retrieval | CQADupstack, Quora | 69.6 | 65.3 | 93.8% |
| Passage Retrieval / General QA | MSMARCO, NFCorpus, NQ | 50.1 | 47.0 | 93.8% |
| Domain-specific QA | HotpotQA, FiQA2018 | 68.4 | 61.1 | 89.3% |
| Entity Retrieval | DBPedia | 48.2 | 42.0 | 87.1% |
| Citation Prediction | SCIDOCS | 23.8 | 20.7 | 87.0% |

lightweight design on code and reasoning-intensive retrieval. These tasks typically require higher reasoning and query-understanding ability, which is not suitable for an extreme lightweight query encoder. Nevertheless, LightRetriever still maintains competitive absolute performance, suggesting promising generalization with room for further task-specific adaptation.

# 4 ABLATIONS

Our core design adopts an asymmetric architecture, with a lightweight query Embedding Bag (Emb) and a full-sized document model for inference. Specifically, we utilize a full-sized model on both sides at training time, then cache all query token embeddings in one Embedding Lookup for fast query-side inference. In ablation studies below (A1-A2), we are interested in other potential symmetric/asymmetric architectures, which are described in the following Table 4.

Table 4: Symmetry Ablation settings (A1-A2). Emb denotes Embedding Bag. 1L-Trans denotes a single-layer Transformer.

| | Train | | Inference | |
|------|-------|-----|-----------|-----|
| | Query | Doc | Query | Doc |
| **Ours** | Full | Full | Emb | Full |
| **A1** | Full | Full | Emb | Emb |
| **A2** | Emb | Full | Emb | Full |

Table 5: Ablations (A1-2) results (nDCG@10) of LightRetriever.

| Ablations | BeIR | CMTEB-R |
|-----------|------|---------|
| Llama-1b | 48.7 | 58.7 |
| A1: Both-side Light | $34.9_{-13.8}$ | $40.1_{-18.6}$ |
| A2: $Enc_q$ use a Emb | $37.5_{-11.2}$ | $41.3_{-17.4}$ |

**A1. Is symmetric lightweight inference effective?**    To test the necessity of above asymmetry, we evaluate a symmetric lightweight setup where both sides use simplified dense encoders. As shown in Table 5, this leads to severe performance drops on both BeIR (–13.8) and CMTEB-R (–18.6), confirming that full document representations are crucial for maintaining retrieval effectiveness. This symmetric setting is not functional for sparse retrieval due to the absence of learnable parameters.

The best symmetric lightweight retriever is Static Embedding (one Embedding Bag) + BM25 (term-based) in Table 2, with nDCG@10 of 44.7 on BeIR and 52.1 on CMTEB-R. However, due to the lack of deep modeling on documents, such a combination is still outperformed. We also tested a reversed asymmetry with deep query encoding and lightweight documents. However, we observed training instability and degraded performance, likely caused by embedding collapse.

**A2. Can we use only a query Embedding Bag for training?**    To confirm that a deep query encoder is still needed during training, we replaced the Transformer-based query encoder with a simple Embedding Bag that reuses only the LLM's embedding layer. This setup significantly degrades performance (–11.2 on BeIR, –17.4 on CMTEB-R), indicating that although deep token interaction could be limited with a durable cost during query inferencing, the full-sized modeling for query side training remains essential to learn effective representations.

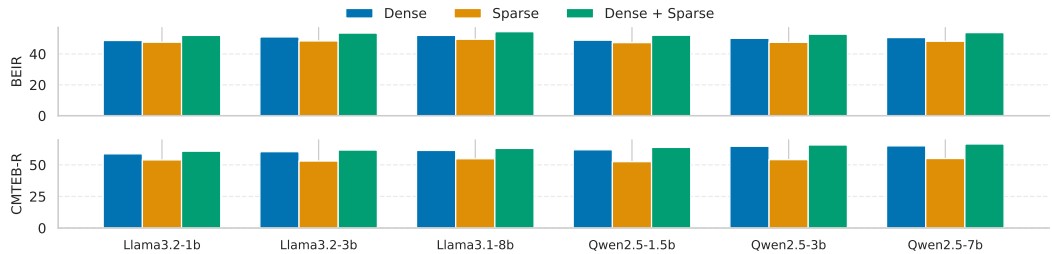

Figure 5: Ablations of retrieval performances (nDCG@10) of LightRetriever on BeIR and CMTEB-R benchmarks. For detailed results, please refer to Table 14, 15 and 16 in Appendix.

**A3. Breakdown Ablation of Retrieval Performance** Figure 5 reports LightRetriever's performance under dense, sparse, and hybrid configurations. Dense retrieval consistently outperforms sparse across all backbones (e.g., LLaMA3, Qwen2.5) and model sizes (1B–8B). While sparse retrieval alone is less effective, the hybrid approach recovers most of the performance gap with minimal overhead. For instance, LLaMA3-1B achieves 52.0 and 60.8 nDCG@10 on BeIR and CMTEB-R in the hybrid setting, with only 1.1 and 3.0 points below the full model, but with a much smaller query encoder. These results confirm the complementarity of both signals and the effectiveness of LightRetriever in balancing efficiency and accuracy.

## 5 CONCLUSION

Existing LLM-based retrievers use symmetric dual-encoders to model both queries and documents. While documents could be pre-encoded and indexed offline, queries arrive in real-time and need online encoding. The deployment of LLMs results in inefficient online services. In this paper, we propose LightRetriever, a novel LLM-based hybrid retrieval architecture capable of extremely lightweight query encoding. Our approach achieves over a 1000x speedup for query inference and 10x increase in end-to-end retrieval throughput on an A800 GPU. Experiments show great robustness and generalization abilities across different foundation LLMs and retrieval tasks.

ACKNOWLEDGMENTS

This work is supported by the National Natural Science Foundation of China (No. U24A20335).

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

# A APPENDIX

## A.1 LIMITATIONS

Our work focuses on online efficiency optimizations of recent LLM-based architectures, whose giant parameters pose a greater need for such optimizations. Other readers may be interested in the application of our methods on non-LLM architectures, e.g., BERT. While such a combination is possible, due to limited time constraints and lesser needs for non-LLM, we leave this to future work. Moreover, as stated in Section §3.2, our query-lightweight design is applicable to normal relevance-based retrieval tasks. Tasks that require complex query understanding, such as reasoning-intensive retrieval, are not suitable for our work.

## A.2 RELATED WORKS

### A.2.1 TEXT RETRIEVAL

Text retrieval builds on top of the vector space model (Salton et al., 1975), which encodes query and document representation vectors, and performs relevance search based on vector similarities.

**Dense retrieval** Dense retrieval (Karpukhin et al., 2020) trains PLM-based encoders to condense queries and passages into low-dimensional dense vectors, then performs relevance search based on the Maximum Inner Product Search (MIPS) (Mussmann & Ermon, 2016) algorithm. In recent years, dense retrievers, such as BGE (Chen et al., 2024) and E5 (Wang et al., 2024a), have gained popularity for strong retrieval abilities across different tasks (Muennighoff et al., 2023; Thakur et al., 2021), languages (Zhang et al., 2023), and context granulaties (Williams et al., 2018; Nguyen et al., 2016), owing to diverse training data, retrieval-customized pre-training (Gao & Callan, 2021; Xiao et al., 2022; Ma et al., 2024), improved negative-mining (Xiong et al., 2021), and so on. Recent SOTA retrievers (Wang et al., 2024b; BehnamGhader et al., 2024; Lee et al., 2024) start to utilize LLMs (Jiang et al., 2023; Touvron et al., 2023) as backbone encoders. With larger parameters and pre-training data, LLM-based retrievers enable significantly powerful retrieval capacities and domain adaptation abilities. However, LLMs also incur heavy online workloads when serving as query encoders. Improving the serving efficiency of LLM-based retrievers is still left for exploration.

**Sparse retrieval** Traditional statistical sparse retrieval algorithms (e.g., TF-IDF (Salton & Buckley, 1988), and BM25 (Robertson et al., 1994)) do not involve learnable language models. Instead, they directly analyze lexical frequencies (e.g., TF) and importance (e.g., IDF) in a large vocabulary space. These lexicals are sparsely distributed and require building inverted indexes (Luhn, 1957) for feasible similarity metric calculation, such as Apache Lucene (Bialecki et al., 2012) and Tantivy. In recent years, sparse retrieval has started to work with pre-trained LMs (PLMs). BGE (Chen et al., 2024), SPLADE (Formal et al., 2021), SparTerm (Bai et al., 2020), and SPARTA (Zhao et al., 2021) use customized XLM-RoBERTa (Conneau et al., 2020) or BERT (Devlin et al., 2019) as backbone encoders to learn term frequencies in an end-to-end manner and directly perform impact search based on the dot product of sparse vectors. Because of sparsity, these works also search with inverted indexes. Recent works (Zhuang et al., 2024) also try to incorporate LLMs with sparse retrieval. However, the online efficiency issue still exists. Learning dynamic analysis (Mackenzie et al., 2023) on learned sparse retrieval shows that semantic information can be fully carried on the document side regardless of token semantics. Motivated by this finding, in this paper we hypothesize that if the semantic capacity is concentrated on the document side, then the query side can be made extremely lightweight with minimal performance degradation.

**Hybrid retrieval** Hybrid retrieval is a technique that combines retrieval scores (Wang et al., 2021) or ranks (Cormack et al., 2009) from multiple retrieval systems, such as dense (Karpukhin et al., 2020) / sparse (Robertson et al., 1994; Formal et al., 2021) retrieval, and single (Karpukhin et al., 2020) / multiple (Khattab & Zaharia, 2020; Chen et al., 2024) vector retrieval. It enables better retrieval abilities by interpolating results from multiple sources. Linear interpolation (Wang et al., 2021) of dense and sparse scores is used in our work.

### A.2.2 Improving inference efficiency

Numerous efforts endeavor to improve the inference efficiency of text retrieval.

**Distilling smaller encoders**  Previous works have tried to distill small query encoders from large encoders for better query inference efficiency. KALE (Campos et al., 2023) prunes the BERT-based query encoder and distills it during post-training. However, when the query encoder is pruned to one Transformers layer, its recall@20 on NQ (Kwiatkowski et al., 2019) decreases by 25-30%. (Wang & Lyu, 2023) proposes asymmetric dual-encoders by pruning BERT layers or using smaller BERT models. Its experiments show that a 2-layer BERT query encoder retains 92.5% of the full-sized performances via distillation and proper student initialization. USTAD (Kim et al., 2024) also proposes similar asymmetry and distillation strategies, which choose the 12-layer BERT as teachers and the 6-layer DistilBERT or 2-layer BERT-mini as students. Its experiments show that the distilled BERT-mini query encoder achieves 95-97% of the teacher's performance because of better distillation.

However, these studies have primarily focused on small language models (SLMs), such as BERT (Devlin et al., 2019), and overlooked feasible approaches for improving online inference efficiency of LLM-based retrievers, where such optimization is even more critical. Furthermore, these existing methods typically retain Transformer layers in the query encoders, which limits their efficiency gains. In addition, most of these works rely on SLMs and are trained on relatively narrow datasets, such as MS-MARCO (Nguyen et al., 2016), thus unable to compete with LLM-based retrievers.

In contrast, with the rapid advancement of LLM-based text retrievers, driven by larger and more diverse foundation models and datasets, retrieval performance has significantly improved. To address the growing need for efficient online inference in this setting, we propose a novel method that, to our knowledge, is the first to target the online efficiency challenge in LLM-based retrievers. Our approach eliminates the need for Transformers layers for query encoders and does not rely on mandatory distillation. Yet our work only incurs an average performance drop of about 5%. Moreover, we evaluate our method across multiple LLMs of varying sizes and diverse retrieval tasks, demonstrating strong generalization capabilities.

**Shrinking index sizes**  Previous works have also been dedicated to reducing index sizes for smaller disk space usage. Matryoshka representation learning (MRL) (Kusupati et al., 2022) supports adjusting the dense vector dimensions by directly taking the top-k dimension of the original vector as shrink vectors. It aligns and trains the corresponding multi-dimensional vectors in the end-to-end contrastive learning. For example, if $k \in \{64, 128, 256\}$, query-passage dense vectors with $\{64, 128, 256\}$-dims could be obtained together after one encoder forward operation. For LM-based sparse retrieval, the similarity scores are directly affected by the impacts of corresponding lexicals (Formal et al., 2021). Thus, it naturally supports reducing the index sizes by retaining Top-k terms/lexicals. To explore the potential of controlling index sizes, our work also explores shrinking index sizes with dense MRL and sparse Top-k in the Section A.4, which also shows strong generalization capacities.

### A.3 Case Study

Here, we report some hand-crafted ranking cases to show LightRetriever's ability to understand complex queries rather than simple lexical matches, even if the query encoder is simplified and limited for inference efficiency.

**Case Study 1: Distinguish compositional semantics.**  This case tests the ability to search multiple attributes together, capturing *fast-moving subjects* with *minimal motion blur*. While both candidate documents discuss camera capabilities, only Doc A contains the correct composition of features. Doc B, although relevant to photography, emphasizes low-light and landscape use cases, which do not align with the compositional semantics. LightRetriever, despite using a simplified query encoder, successfully prioritizes Doc A across dense, sparse, and hybrid variants.

**Case Study 2:  Negation.**  This case tests whether models can correctly interpret affirmation/negation. The query explicitly seeks articles *against* remote work for improving collaboration.

Table 6: Case Study 1: Distinguish compositional semantics. Texts colored in red are important for semantic matching, while the blue ones represent unrelated words or distractions. ✓ means ranking correct, while × means rank fail.

| Query | Which camera captures fast-moving subjects with minimal motion blur? |
|---|---|
| **Doc A (relevant)** | The X200 features a 1/8000s shutter and phase-detection AF, enabling crisp photos of sports and fast action with minimal motion blur. |
| **Doc B** | The Y7 is praised for its wide-angle lens and superior low-light performance, often used for landscapes and night photography. |
| **Results** | Full-sized Llama8b ✓ **Doc A** > Doc B
LightRetriever-Llama8b$_{Dense}$ ✓ **Doc A** > Doc B
LightRetriever-Llama8b$_{Sparse}$ ✓ **Doc A** > Doc B
LightRetriever-Llama8b$_{Hybrid}$ ✓ **Doc A** > Doc B |

Table 7: Case Study 2: Negation.

| Query | Articles arguing against remote work for improving team collaboration. |
|---|---|
| **Doc A (relevant)** | A recent study argues that remote work hinders spontaneous collaboration, noting dropoffs in serendipitous conversations and mentoring opportunities among junior staff. |
| **Doc B** | Many experts advocate remote work for flexibility and increased individual productivity, highlighting asynchronous tools that maintain collaboration. |
| **Results** | Full-sized Llama8b ✓ **Doc A** > Doc B
LightRetriever-Llama8b$_{Dense}$ ✓ **Doc A** > Doc B
LightRetriever-Llama8b$_{Sparse}$ ✓ **Doc A** > Doc B
LightRetriever-Llama8b$_{Hybrid}$ ✓ **Doc A** > Doc B |

Although both candidate documents discuss remote work and collaboration, only Doc A aligns with the negative phase by describing how remote work hinders interactions. Doc B, on the contrary, advocates for remote work, which is semantically opposite to the query intent. Correctly distinguishing the affirmation/negation requires more than lexical overlap with *remote work* and *collaboration*. As shown in the results, both the full-sized Llama8b and all LightRetriever variants successfully rank Doc A higher, demonstrating that LightRetriever can preserve such negation discrimination ability.

Table 8: Case Study 3: Complex understanding.

| Query | How to revert the last Git commit without losing the changes? |
|---|---|
| **Doc A (relevant)** | Use git reset –soft HEAD 1 to undo the last commit while keeping changes in the working tree and index; this retains your edited files for further edits. |
| **Doc B** | Use git revert HEAD to create a new commit that undoes the changes introduced by the previous commit, preserving history and not altering the working tree. |
| **Results** | Full-sized Llama8b ✓ **Doc A** > Doc B
LightRetriever-Llama8b$_{Dense}$ × **Doc A** < Doc B
LightRetriever-Llama8b$_{Sparse}$ ✓ **Doc A** > Doc B
LightRetriever-Llama8b$_{Hybrid}$ × **Doc A** < Doc B |

**Case Study 3: Complex understanding.** This case tests the ability to understand complex technical instructions. The query asks how to revert the last Git commit *without losing the changes*, a subtle requirement that distinguishes between keeping edits in the working tree (Doc A) versus creating a new commit that reverts prior changes (Doc B). Despite both documents mentioning ways to undo commits, only Doc A satisfies the instructions. The full-sized Llama8b is able to complete such complex technical instructions. However, LightRetriever-Dense and Hybrid fail, likely because they focus on the strong lexical signal of *revert* but neglect the constraint *without losing the changes*. Interestingly, LightRetriever-Sparse ranks Doc A correctly, suggesting that token-level sparse matching may emphasize critical keywords such as *keeping changes* and *working tree*. This

highlights a limitation of lightweight dense encoding: complex compositional understanding may fail, and success may depend on complementary sparse signals.

## A.4 ADDITIONAL RESULTS ABOUT CONTROLS OF VECTOR DIMENSION, SPARSITY, AND EMBEDDING SERVING SIZE

Retrieval systems usually involve indexing millions of documents into search databases, which consumes large amounts of disk space, especially for dense vectors. For example, indexing 8.8M MS-MARCO passages (Nguyen et al., 2016) with the Llama-8b model consumes 134.92 GB for Faiss dense retrieval and 11.24 GB for Lucene sparse retrieval. Besides index sizes, LightRetriever's dense part also serves as an Embedding matrix in RAM with shape [V, H] in Float16 precision. For the Llama-8b model, if the serving size is not properly controlled, such online embedding will consume 1.05 GB of RAM.

Thus, it's essential to explore the possibility of controlling index sizes and serving sizes by shrinking the vector dimensions. 1) For dense retrieval, we incorporate Matryoshka Representation Learning (MRL) (Kusupati et al., 2022) with LightRetriever to support flexible controls of dense embedding dimensions. Given a dense vector $v^{den}$ with dimension $H$, MRL derives dimension-shrunk vectors by simply cutting out the Top-k dimensions $v^{den}[: k]$, then applies the same training loss as the original vectors. MRL trains a collection of Top-k dimensions together, enabling flexible choosing of a proper k. 2) For sparse retrieval, the sparse vector directly learns the term impact (i.e., lexical frequency) within a large sparse dimension. Thus, LightRetriever supports controlling sparsity by directly taking the Top-k terms, where the larger impact terms matter more.

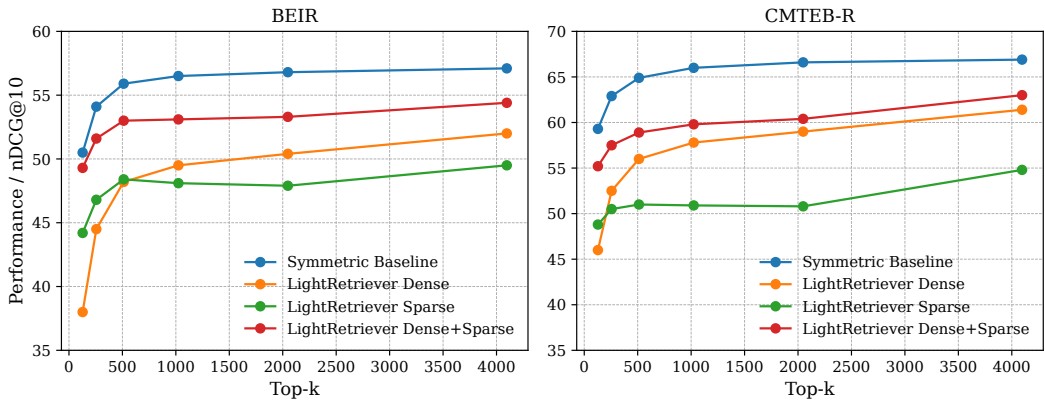

Figure 6: Performances (Llama-3.1-8b) with different Top-k dimensions or sparsities.

Table 9: Performances, dense serving size, and index size of LightRetriever-Llama8b across different top-k dimensions (Dim). Note that dim=4096 is the untruncated setting.

| Dimension | Performance / nDCG@10 | | Serving Size (MB) | Index Size (GB) | |
|---|---|---|---|---|---|
| Top-k | BEIR | CMTEB-R | Embedding | Dense | Sparse |
| 128 | 49.3 | 55.2 | 32.83 | 4.22 | 3.03 |
| 256 | 51.6 | 57.5 | 65.67 | 8.43 | 5.42 |
| 512 | 53.0 | 58.9 | 131.33 | 16.87 | 8.89 |
| 1024 | 53.1 | 59.8 | 262.67 | 33.73 | 10.85 |
| 2048 | 53.3 | 60.4 | 525.34 | 67.46 | 11.22 |
| 4096 | 54.4 | 63.0 | 1050.67 | 134.92 | 11.24 |

**Index size controls via tuning vector dimension and sparsity** We conduct experiments of controlling dense vector dimension and sparse vector sparsity with Top-k of $\{128, 256, 512, 1024, 2048, 4096\}$ on Llama-3.1-8b. As presented in Figure 6 and Table 9, LightRetriever has affordable performance cost when truncating the Top-k. The lowest Top-k setting (128)

has 5.1 and 7.8 nDCG@10 performance gaps on BeIR and CMTEB-R benchmarks compared to the highest Top-k (4096), but its disk usages with MS-MARCO are only 4.22 GB for dense trieval and 3.03 GB for sparse retrieval. This shows that our LightRetriever is capable of flexible index controls via simple vector shrinking techniques, with limited effect on retrieval performances.

**Dense embedding serving size controls via dimension truncation** By applying dimension truncation with MRL, LightRetriever also allows serving embeddings at flexible sizes proportional to the selected Top-k dimensions. As is shown in Table 9, truncating from 4096 to 1024 dimensions reduces the online embedding size from 1.05 GB to only 262 MB, a $4\times$ reduction with affordable performance loss (53.1 vs. 54.4 nDCG@10 on BEIR). Under the extreme truncation to 128 dimensions, the serving size is reduced to just 32 MB, making it practical to deploy LightRetriever on resource-constrained environments such as memory-limited servers. This demonstrates that LightRetriever supports cost-effective online serving with controllable degradation in retrieval quality, offering a scalable solution for balancing efficiency and effectiveness in real-world deployments.

## A.5 Additional ablation about the effect of auxiliary KL loss

Table 10: Ablations (nDCG@10, w/Llama-3.1-8b) on BeIR and CMTEB-R about auxiliary KL loss.

| Benchmark | w/ KL | | wo/ KL | |
|---|---|---|---|---|
| | BeIR | CMTEB-R | BeIR | CMTEB-R |
| Dense | 52.0 | 61.4 | $50.8_{-1.2}$ | $60.1_{-1.3}$ |
| Sparse | 49.5 | 54.8 | $48.7_{-0.8}$ | $54.4_{-0.4}$ |
| Hybrid | 54.4 | 63.0 | $53.9_{-0.5}$ | $62.1_{-0.9}$ |

Previous works for BERT-based asymmetric retrievers (Wang & Lyu, 2023; Kim et al., 2024) emphasize the importance of alignment between asymmetric students and full symmetric teachers. To explore the effect of such alignment, besides contrastive loss as the main loss function, we also explore to adopt the KL loss as an auxiliary to align the asymmetric similarity scores $S^{student} = v_q^{den} \cdot v_d^{den}$ to full-sized symmetric similarity scores $S^{teacher} = Enc_q(q) \cdot Enc_d(d)$,

$$\ell^{Align} = \texttt{KL-Div}(S^{student}, S^{teacher}) \tag{8}$$

Results in Table 10 show that removing such loss incurs limited drops of -0.5 and -0.9 on BeIR and CMTEB-R. Thus, such auxiliary KL loss is not mandatory needed in our experiments.

Table 11: Training Dataset informations.

| Dataset | Language | Category | Deduped Size | Epoch | Ratio |
|---|---|---|---|---|---|
| Amazon Review (2018) (Ni et al., 2019) | English | Amazon | 999999 | 0.01 | 0.58% |
| cMedQA2 (Zhang et al., 2018) | Chinese | Chinese Medical | 99936 | 1 | 5.84% |
| DuReader (Qiu et al., 2022) | Chinese | Chinese Web Collections | 86366 | 1 | 5.05% |
| Eli5 (Fan et al., 2019) | English | Reddit | 325390 | 0.05 | 0.95% |
| Fever (Thorne et al., 2018) | English | Wikipedia QA | 109808 | 0.8 | 5.13% |
| FiQA (Maia et al., 2018) | English | Financial | 5498 | 1 | 0.32% |
| GooAQ Pairs (Khashabi et al., 2021) | English | Web Collections | 3012347 | 0.025 | 4.40% |
| HotpotQA (Yang et al., 2018b) | English | Wikipedia QA | 85000 | 0.5 | 2.48% |
| MSMARCO (Nguyen et al., 2016) | English | Web Collections | 502854 | 1 | 29.39% |
| NFCorpus (Boteva et al., 2016) | English | Medical | 2585 | 1 | 0.15% |
| NPR (Lucas et al., 2023) | English | News | 594376 | 0.05 | 1.74% |
| NQ (Kwiatkowski et al., 2019) | English | Wikipedia QA | 58800 | 1 | 3.44% |
| PQA Pairs (Lewis et al., 2021) | English | Wikipedia QA | 99999 | 0.5 | 2.92% |
| Quora Duplicates Triples (Iyer et al., 2017) | English | Forum Duplicates | 97011 | 0.5 | 2.83% |
| S2ORC Title-Abstract (Lo et al., 2020) | English | Semantic Scholar | 99998 | 0.5 | 2.92% |
| SciFact (Wadden et al., 2020) | English | S2ORC | 806 | 1 | 0.05% |
| SPECTER (Cohan et al., 2020) | English | Semantic Scholar | 136642 | 0.25 | 2.00% |
| StackExchange Dups (Title-Body) (da Silva et al., 2018) | English | Forum Duplicates | 250516 | 0.25 | 3.66% |
| E5 Synthetic (BeastyZ, 2024) | English | GPT Synthetic Data | 224791 | 1 | 13.14% |
| T2Ranking (Xie et al., 2023) | Chinese | Chinese Web Collections | 200362 | 0.5 | 5.86% |
| Trivia (Joshi et al., 2017) | English | Wikipedia QA | 60370 | 0.5 | 1.76% |
| Wikihow (Koupaee & Wang, 2018) | English | WikiHow | 128543 | 0.25 | 1.88% |
| Yahoo Answers (Title-Answer) (Zhang et al., 2015) | English | Yahoo | 1198018 | 0.05 | 3.50% |

A.6    TRAINING DATASETS

**Dataset infos**    Following the settings of previous LLM-based retrievers (Wang et al., 2024b; BehnamGhader et al., 2024; Ma et al., 2025), our work uses 23 training sets pre-built by tDRO (Ma et al., 2025) or Sentence Transformers Training Data, including 20 English and 3 Chinese datasets. Detailed dataset information is listed in Table 11 as follows.

Many downstream retrieval benchmarks (Thakur et al., 2021; Xiao et al., 2024) are evaluated out-of-domain, where no training samples from the downstream sources are available. To avoid the overfitting of training sets, as is shown in the above Table 11, we slightly tune the epoch used in the dataset sampling, by following previous works in (Ni et al., 2022; Wang et al., 2024b; Ma et al., 2025).

Table 12: Training set deduplications. We report each dataset with the original size, deduplicated size, and number of duplicates detected in both the train and test sets.

| Dataset | Original Size | Deduped Size | Duplicates |
|---|---|---|---|
| Amazon Review (2018) | 1000000 | 999999 | 1 |
| cMedQA2 | 100000 | 99936 | 64 |
| DuReader | 86395 | 86366 | 29 |
| Eli5 | 325475 | 325390 | 85 |
| Fever | 109810 | 109808 | 2 |
| FiQA | 5498 | 5498 | 0 |
| GooAQ Pairs | 3012496 | 3012347 | 149 |
| HotpotQA | 85000 | 85000 | 0 |
| MSMARCO | 502939 | 502854 | 85 |
| NFCorpus | 2590 | 2585 | 5 |
| NPR | 594384 | 594376 | 8 |
| NQ | 58812 | 58800 | 12 |
| PQA Pairs | 100000 | 99999 | 1 |
| Quora Duplicates Triples | 101762 | 97011 | 4751 |
| S2ORC Title-Abstract | 100000 | 99998 | 2 |
| SciFact | 809 | 806 | 3 |
| SPECTER | 136645 | 136642 | 3 |
| StackExchange Duplicates (Title-Body) | 250519 | 250516 | 3 |
| E5 Synthetic | 224791 | 224791 | 0 |
| T2Ranking | 200376 | 200362 | 14 |
| Trivia | 60380 | 60370 | 10 |
| Wikihow | 128543 | 128543 | 0 |
| Yahoo Answers (Title-Answer) | 1198260 | 1198018 | 242 |

Table 13: Samples of duplicate queries found in both the train and test sets.

| Train Set | Train qid | Train Query | Test Set | Test qid | Test Query |
|---|---|---|---|---|---|
| Quora | train-164 | Is there a framework for auditing social media? | Quora | 201573 | Is there a framework for auditing social media? |
| Yahoo-Answer | train-10046 | who is sachin tendulkar? | Quora | 47702 | Who is Sachin Tendulkar? |
| MSMARCO | 968274 | where did abraham lincoln died | DBPedia | QALD2_tr-6 | Where did Abraham Lincoln die? |
| Eli5 | train-3088 | How does unemployment insurance work? | FiQA | 2648 | How does unemployment insurance work? |

**Training set decontamination**    Training set decontamination is essential for developing fair, unbiased, and robust retrieval systems. Existing training datasets are derived from multiple sources, whose training data have possible duplicates with the test sets. Following tDRO (Ma et al., 2025),

we perform strict training set decontamination with SimHash (Manku et al., 2007) to ensure no training queries appear in the BeIR (Thakur et al., 2021) and CMTEB Retrieval (Xiao et al., 2024). The original sizes, deduplicated sizes, and duplicate numbers as listed in the Table 12.

Additionally, we also listed some samples of the duplicated training queries, as well as the corresponding test sets in Table 13.

### A.7 TASK INSTRUCTIONS

As previously mentioned in Section 2.1, modern LLM-based retrievers (Wang et al., 2024b; BehnamGhader et al., 2024) introduce task-specific instructions to format input queries, which enables higher retrieval capacities. These LLM-based instruction-tuned retrieval is similar to instruction-based supervised fine-tuning (SFT) in LLM generation (Ouyang et al., 2022). Following Mistral-E5 (Wang et al., 2024b), we reuse most of its task instructions, and format the instruction and query in the following format:

$$\text{Instruct: } instruction \backslash \text{nQuery: } query$$

The detailed instructions are listed below, most of which are directly copied from the previous works E5-Mistral (Wang et al., 2024b) and tDRO (Ma et al., 2025) for reproducibility. Note that one training set may correspond to multiple instructions to avoid overfitting.

**Instructions for training sets**

1. **Amazon Review (2018)**(1): Given a title, retrieve the corresponding reviews from Amazon

2. **Amazon Review (2018)**(2): Given a title, retrieve a Amazon review

3. **cMedQA2**: Given a Chinese community medical question, retrieve replies that best answer the question

4. **DuReader**: Given a Chinese search query, retrieve web passages that answer the question

5. **Eli5**: Provided a user question, retrieve the highest voted answers on Reddit ELI5 forum

6. **GooAQ Pairs**: Given a web search query, retrieve the corresponding answers from Google

7. **HotpotQA**: Given a multi-hop question, retrieve documents that can help answer the question

8. **MSMARCO**: Given a web search query, retrieve relevant passages that answer the query

9. **NQ**(1): Given a question, retrieve Wikipedia passages that answer the question

10. **NQ**(2): Retrieve Wikipedia passages that answer the question

11. **Quora Duplicates Triples**(1): Given a question, retrieve questions that are semantically equivalent to the given question

12. **Quora Duplicates Triples**(2): Find questions that have the same meaning as the input question

13. **S2ORC Title-Abstract**(1): Given a title, retrieve the abstract from scientific papers

14. **S2ORC Title-Abstract**(2): Given a title, retrieve abstracts from scientific papers that match the title

15. **SciFact**: Given a scientific claim, judge whether the document supports or refutes the claim

16. **SPECTER**(1): Given a title, retrieve semantic related titles

17. **SPECTER**(2): Retrieve semantic related titles from scientific publications

18. **StackExchange Duplicates (Title-Body)**: Retrieve duplicate questions and passages from StackOverflow forum

19. **T2Ranking**: Given a Chinese search query, retrieve web passages that answer the question

20. **Trivia**(1): Given a question, retrieve Wikipedia passages that answer the question

21. **Trivia**(2): Retrieve Wikipedia passages that answer the question

22. **Wikihow**: Given a summary, retrieve Wikipedia passages that match the summary

23. **Yahoo Answers (Title-Answer)**: Given a title, retrieve Yahoo answers that match the title

**Instruction for English BeIR benchmarks**

1. **ArguAna**: Given a claim, find documents that refute the claim

2. **ClimateFEVER**: Given a claim about climate change, retrieve documents that support or refute the claim

3. **CQADupStack**: Given a question, retrieve detailed question descriptions from Stackexchange that are duplicates to the given question

4. **DBPedia**: Given a query, retrieve relevant entity descriptions from DBPedia

5. **FEVER**: Given a claim, retrieve documents that support or refute the claim

6. **FiQA2018**: Given a financial question, retrieve user replies that best answer the question

7. **HotpotQA**: Given a multi-hop question, retrieve documents that can help answer the question

8. **MSMARCO**: Given a web search query, retrieve relevant passages that answer the query

9. **NFCorpus**: Given a question, retrieve relevant documents that best answer the question

10. **NQ**: Given a question, retrieve Wikipedia passages that answer the question

11. **Quora**: Given a question, retrieve questions that are semantically equivalent to the given question

12. **SCIDOCS**: Given a scientific paper title, retrieve paper abstracts that are cited by the given paper

13. **SciFact**: Given a scientific claim, retrieve documents that support or refute the claim

14. **Touche2020**: Given a question, retrieve detailed and persuasive arguments that answer the question

15. **TRECCOVID**: Given a query on COVID-19, retrieve documents that answer the query

**Instruction for Chinese CMTEB Retrieval benchmarks**

1. **CmedqaRetrieval**: Given a Chinese community medical question, retrieve replies that best answer the question

2. **CovidRetrieval**: Given a question on COVID-19, retrieve news articles that answer the question

3. **DuRetrieval**: Given a Chinese search query, retrieve web passages that answer the question

4. **EcomRetrieval**: Given a user query from an e-commerce website, retrieve description sentences of relevant products

5. **MMarcoRetrieval**: Given a web search query, retrieve relevant passages that answer the query

6. **MedicalRetrieval**: Given a medical question, retrieve user replies that best answer the question

7. **T2Retrieval**: Given a Chinese search query, retrieve web passages that answer the question

8. **VideoRetrieval**: Given a video search query, retrieve the titles of relevant videos

A.8 EFFICIENCY OPTIMIZATIONS TECHNIQUES

**Customized causal mask**  As defined in the Equation 3, our work computes dense token vectors for each input token with a common prompt. This causes the repeated computation of the common prompt area. We develop a customized causal mask for LLMs to avoid such computation waste. Specifically, given a task instruction $Inst$ and query tokens $T_q = \{t_0, t_1, ..., t_{n-1}\}$, we format the instruction and query tokens as follows,

$$\text{Input IDs} = \texttt{<bos>}\ Inst\ t_0\ \texttt{<eos>}\ \ldots\ t_{n-1}\ \texttt{<eos>} \tag{9}$$

The customized causal mask ensures that the token blocks can attend to the common prompt area, but not attend to each other. Let the sequence length be $L$, prompt length be $P$, micro block (each token + eos) width be $w$, and let $q \in [0, L-1]$ denote the query index, $k \in [0, L-1]$ the key index. The attention mask (0 means attend, $-\infty$ means masked) is defined as,

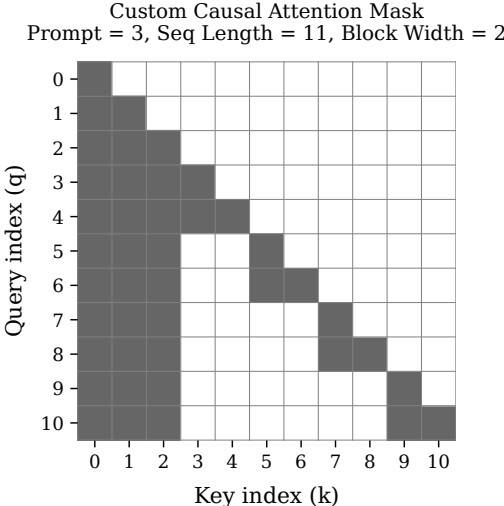

Figure 7: An example of the customized attention mask.

$$\text{Mask}[q,k] = \begin{cases} 0, & \text{if } q < P \text{ and } k \leq q \quad \text{(attn within prompt)} \\ 0, & \text{if } q \geq P \text{ and } k < P \quad \text{(blocks can attend to prompt)} \\ 0, & \text{if } q \geq P \text{ and } \left\lfloor \frac{q-P}{w} \right\rfloor = \left\lfloor \frac{k-P}{w} \right\rfloor \text{ and } k \leq q \quad \text{(attn within current block)} \\ -\infty, & \text{otherwise} \end{cases}$$

An example of the customized causal mask is shown in Figure 7.

**Optimizations of sparse max aggregation** As stated in Equation 6, the sparse vector is computed via

$$\text{1. Project to vocab with dim } V \rightarrow \text{2. ReLU} \rightarrow \text{3. Log Saturation} \rightarrow \text{4. Max Aggregation} \quad (10)$$

, where the vocabulary dimension $V$ is much larger than the hidden dimension $H$. For example, for Llama3.1-8b (Touvron et al., 2023) model, $V = 128256$ and $H = 4096$, where $\lfloor V/H \rfloor = 31$. This means that, given batch size B, sequence length T, hidden size H, vocabulary size V, the intermediate tensors with shape $[B, T, V]$ in the projection, ReLU, and Log are 31x larger than a dense tensor $[B, T, H]$, which consumes huge amounts of GPU memory during training and inference. However, only a tensor with shape $[B, V]$ is needed for the final aggregated representations. Thus, an optimization for computing the sparse vector is needed urgently.

We design an efficient way of computing the sparse vector. Firstly, because ReLU and Log are monotonically increasing functions, we first move the last max aggregation operations to step 2, right after the projection.

$$\text{1. Project to vocab with dim } V \rightarrow \text{2. Max Aggregation} \rightarrow \text{3. ReLU} \rightarrow \text{4. Log Saturation} \quad (11)$$

Then we design a customized PyTorch Autograd function to fuse steps 1&2. It computes the projection and maximum operations together along the sequence length dimension $T$. Such sliced computation avoids the creation of a large tensor with a sequence length dimension. Our preliminary observation shows that this helps save the GPU memory usage of the sparse max aggregation up to 100-200x. The forward and backward are presented in the Algorithm 1.

---

**Algorithm 1** Forward and Backward Pass of Efficient Projection then Max aggregation

---

**Require:** Input tensor $X \in \mathbb{R}^{B \times T \times H}$, weight $W \in \mathbb{R}^{H \times V}$, optional bias $b \in \mathbb{R}^V$, optional input attention mask $M \in \{0, 1\}^{B \times T}$, where B is batch size, T is sequence length, H is hidden size, V is vocabulary size.
**Ensure:** Output tensor $Y \in \mathbb{R}^{B \times V}$
 1: **function** FORWARD($X, W, b, M$)
 2:     Initialize $Y \leftarrow -\infty$
 3:     **for** $t = 0$ **to** $T - 1$ **do**
 4:         $L_t \leftarrow X[:, t, :] \cdot W$
 5:         **if** $b$ is not None **then**
 6:             $L_t \leftarrow L_t + b$
 7:         **end if**
 8:         **if** $M$ is not None **then**
 9:             **for** $i = 0$ **to** $B - 1$ **do**
10:                 **if** $M[i, t] = 0$ **then**
11:                     $L_t[i, :] \leftarrow -\infty$
12:                 **end if**
13:             **end for**
14:         **end if**
15:         $Y \leftarrow \max(Y, L_t)$
16:     **end for**
17:     Save argmax$_t$ indices as `max_indices`
18:     **return** $Y$
19: **end function**
20: **function** BACKWARD($\nabla Y,$ `max_indices`$, X, W$)
21:     Initialize $\nabla X \leftarrow 0, \nabla W \leftarrow 0, \nabla b \leftarrow 0$
22:     **for** $t = 0$ **to** $T - 1$ **do**
23:         $X_t \leftarrow X[:, t, :] \in \mathbb{R}^{B \times H}$
24:         $\nabla X_t \leftarrow \nabla X[:, t, :] \in \mathbb{R}^{B \times H}$
25:         $M_t \leftarrow$ `max_indices`$_{I==t} \in \{0, 1\}^{B \times V}$
26:         $\nabla L_t \leftarrow \nabla Y \odot M_t \in \mathbb{R}^{B \times V}$                          ▷ Only keep grads where $t$ was max
27:         $\nabla X_t += \nabla L_t \cdot W^\top$
28:         $\nabla W += X_t^\top \cdot \nabla L_t$
29:         $\nabla b += \sum_{i=1}^{B} \nabla L_t[i, :]$
30:     **end for**
31:     **return** $\nabla X, \nabla W, \nabla b$
32: **end function**

---

**Other important optimizations** Proper optimizations are important. For example, we found it hard to reproduce quickly with some of the open-sourced baselines (BehnamGhader et al., 2024), which is roughly 6x slower than our implementation when testing with the same size of LLM. Our experiments involve training with 23 train sets and inference with another 23 test sets, which requires proper optimizations urgently. We import other optimizations based on the implementations released by tDRO (Ma et al., 2025). These optimizations include 1) A distributed multi-node inference framework based on PyTorch RPC (Li et al., 2023); 2) Sequence packing (Kundu et al., 2024) with Flash Attention 2 (Dao, 2024), which gives roughly 50% training speedup due to the removal of the pad tokens.; 3) Fused Kernels from Liger-Kernel (Hsu et al., 2024), which gives roughly 10% speedup to both training and inferencing; 4) Multi-vector gradient accumulation support with GradCache (Gao et al., 2021) for optionally enlarging batch sizes if necassary.

## A.9 DETAILED MAIN RESULTS

The detailed main results (nDCG@10) on each dataset are shown in the tables below, which are full versions of Table 2.

Table 14: (Part1) Detailed main results (nDCG@10) on BeIR test sets (except MSMARCO, it uses the dev set).

| Task / Test, nDCG@10 | ArguAna | CQADup | CFEVER | DBPedia | FEVER | FiQA | HotpotQA | MSMARCO |
|---|---|---|---|---|---|---|---|---|
| Static Embedding | 44.4 | 22.0 | 20.4 | 27.4 | 43.0 | 20.0 | 46.8 | 17.9 |
| BM25 | 31.5 | 29.9 | 21.3 | 31.3 | 75.3 | 23.6 | 60.3 | 22.8 |
| USTAD | 34.9 | 30.6 | 22.5 | 35.9 | 76.9 | 29.5 | 56.0 | 46.6 |
| BGE-m3-dense | 53.9 | 38.4 | 26.5 | 38.5 | 77.2 | 40.8 | 68.6 | 37.6 |
| BGE-m3-sparse | 36.2 | 30.5 | 26.7 | 24.4 | 85.9 | 26.9 | 70.3 | 17.1 |
| SPLADE-v3 | 50.9 | 34.7 | 23.3 | 45.0 | 79.6 | 37.4 | 69.2 | 46.8 |
| LLM2Vec-llama3-8b | 62.8 | 48.3 | 34.3 | 48.3 | 90.2 | 55.3 | 71.8 | 43.2 |
| E5-Mistral | 61.9 | 42.9 | 38.3 | 48.9 | 87.8 | 56.6 | 75.7 | 43.1 |
| **Llama3.2-1b** | | | | | | | | |
| Full Symmetric | 58.0 | 44.0 | 41.2 | 43.2 | 89.2 | 45.5 | 69.3 | 42.7 |
| LightRetriever Dense | 49.2 | 38.1 | 37.9 | 34.2 | 83.6 | 40.4 | 60.8 | 39.2 |
| LightRetriever Sparse | 49.5 | 34.5 | 27.5 | 35.9 | 84.7 | 35.7 | 65.5 | 37.2 |
| LightRetriever Hybrid | 54.1 | 39.7 | 39.3 | 39.5 | 87.3 | 42.1 | 68.2 | 41.1 |
| **Llama3.2-3b** | | | | | | | | |
| Full Symmetric | 58.5 | 46.8 | 43.7 | 46.4 | 89.9 | 53.4 | 73.7 | 44.3 |
| LightRetriever Dense | 55.3 | 40.3 | 41.7 | 36.5 | 84.0 | 46.2 | 64.3 | 41.6 |
| LightRetriever Sparse | 46.5 | 34.1 | 26.9 | 38.5 | 86.2 | 38.9 | 66.3 | 39.2 |
| LightRetriever Hybrid | 56.3 | 41.4 | 42.4 | 41.6 | 87.6 | 46.9 | 69.7 | 42.9 |
| **Llama3.1-8b** | | | | | | | | |
| Full Symmetric | 57.1 | 49.2 | 44.1 | 48.2 | 90.9 | 59.0 | 77.8 | 46.2 |
| LightRetriever Dense | 55.8 | 40.9 | 41.5 | 38.7 | 85.7 | 49.8 | 67.6 | 43.4 |
| LightRetriever Sparse | 46.4 | 33.6 | 32.2 | 38.8 | 86.1 | 41.9 | 67.9 | 41.0 |
| LightRetriever Hybrid | 57.2 | 41.9 | 42.4 | 42.0 | 87.8 | 50.7 | 71.5 | 44.5 |
| **Mistral0.3-7b** | | | | | | | | |
| Full Symmetric | 56.2 | 49.7 | 43.6 | 48.4 | 91.3 | 59.6 | 77.7 | 46.6 |
| LightRetriever Dense | 53.5 | 41.6 | 40.9 | 38.9 | 85.6 | 50.9 | 66.9 | 44.2 |
| LightRetriever Sparse | 44.0 | 33.6 | 21.9 | 37.0 | 81.0 | 43.6 | 65.4 | 40.9 |
| LightRetriever Hybrid | 54.3 | 42.2 | 41.4 | 40.5 | 86.9 | 51.7 | 69.9 | 44.8 |
| **Qwen2.5-1.5b** | | | | | | | | |
| Full Symmetric | 56.6 | 44.1 | 40.4 | 43.9 | 88.9 | 47.9 | 68.1 | 42.6 |
| LightRetriever Dense | 51.2 | 38.3 | 36.1 | 33.0 | 82.1 | 41.2 | 59.0 | 39.1 |
| LightRetriever Sparse | 46.5 | 35.0 | 25.7 | 36.3 | 85.1 | 35.3 | 63.3 | 37.3 |
| LightRetriever Hybrid | 54.1 | 40.1 | 37.7 | 39.5 | 87.0 | 42.1 | 66.5 | 40.8 |
| **Qwen2.5-3b** | | | | | | | | |
| Full Symmetric | 55.5 | 47.0 | 44.1 | 45.8 | 89.6 | 52.9 | 71.6 | 43.7 |
| LightRetriever Dense | 49.8 | 39.6 | 41.1 | 35.0 | 83.9 | 45.4 | 62.8 | 41.0 |
| LightRetriever Sparse | 45.5 | 36.2 | 28.7 | 36.9 | 84.9 | 38.6 | 64.2 | 38.7 |
| LightRetriever Hybrid | 52.8 | 41.4 | 42.1 | 39.8 | 86.9 | 46.1 | 68.2 | 42.3 |
| **Qwen2.5-7b** | | | | | | | | |
| Full Symmetric | 57.1 | 48.2 | 43.5 | 47.9 | 90.7 | 56.5 | 74.7 | 44.8 |
| LightRetriever Dense | 53.3 | 40.0 | 39.1 | 36.8 | 85.1 | 48.8 | 64.4 | 41.8 |
| LightRetriever Sparse | 47.5 | 35.6 | 28.2 | 36.7 | 85.4 | 40.2 | 65.2 | 40.1 |
| LightRetriever Hybrid | 54.4 | 41.6 | 42.2 | 41.0 | 87.9 | 49.7 | 69.4 | 43.3 |

Table 15: (Part2) Detailed main results (nDCG@10) on BeIR test sets.

| Task / Test, nDCG@10 | NFCorpus | NQ | Quora | SCIDOCS | SciFact | TRECCOVID | Touche2020 | Avg (# 15) |
|---|---|---|---|---|---|---|---|---|
| Static Embedding | 30.0 | 23.1 | 77.5 | 13.2 | 59.3 | 44.6 | 22.4 | 34.1 |
| BM25 | 32.5 | 32.9 | 78.9 | 15.8 | 66.5 | 65.6 | 36.7 | 41.7 |
| USTAD | 30.7 | 50.8 | 81.4 | 14.4 | 55.5 | 72.3 | 24.7 | 44.2 |
| BGE-m3-dense | 31.5 | 60.1 | 88.4 | 15.2 | 63.6 | 47.0 | 20.6 | 47.2 |
| BGE-m3-sparse | 28.3 | 20.3 | 73.4 | 12.2 | 63.3 | 52.7 | 27.8 | 39.7 |
| SPLADE-v3 | 35.7 | 58.6 | 81.4 | 15.8 | 71.0 | 74.8 | 29.3 | 50.2 |
| LLM2Vec-llama3-8b | 41.8 | 64.2 | 87.2 | 23.0 | 78.2 | 80.3 | 20.5 | 56.6 |
| E5-Mistral | 38.6 | 63.5 | 89.6 | 16.3 | 76.4 | 87.2 | 26.4 | 56.9 |
| **Llama3.2-1b** | | | | | | | | |
| Full Symmetric | 35.5 | 58.6 | 89.4 | 19.8 | 72.3 | 66.7 | 20.9 | 53.1 |
| LightRetriever Dense | 30.1 | 52.2 | 87.0 | 17.2 | 66.5 | 68.8 | 24.7 | 48.7 |
| LightRetriever Sparse | 34.0 | 52.0 | 82.2 | 17.2 | 69.1 | 60.6 | 27.8 | 47.6 |
| LightRetriever Hybrid | 34.2 | 56.6 | 87.4 | 18.6 | 71.5 | 70.5 | 30.2 | 52.0 |
| **Llama3.2-3b** | | | | | | | | |
| Full Symmetric | 37.6 | 62.8 | 89.5 | 22.5 | 75.3 | 68.7 | 21.2 | 55.6 |
| LightRetriever Dense | 31.7 | 56.4 | 87.8 | 19.0 | 70.4 | 68.6 | 21.6 | 51.0 |
| LightRetriever Sparse | 33.3 | 55.3 | 83.2 | 17.7 | 67.3 | 64.1 | 28.5 | 48.4 |
| LightRetriever Hybrid | 34.2 | 59.8 | 88.1 | 20.0 | 71.4 | 70.9 | 29.5 | 53.5 |
| **Llama3.1-8b** | | | | | | | | |
| Full Symmetric | 38.6 | 65.5 | 89.9 | 23.8 | 77.3 | 62.0 | 22.2 | 56.8 |
| LightRetriever Dense | 31.8 | 59.0 | 88.5 | 19.3 | 70.6 | 65.7 | 21.1 | 52.0 |
| LightRetriever Sparse | 33.6 | 58.4 | 84.4 | 18.9 | 69.7 | 63.0 | 27.1 | 49.5 |
| LightRetriever Hybrid | 34.6 | 61.9 | 88.6 | 20.7 | 72.6 | 71.5 | 27.8 | 54.4 |
| **Mistral0.3-7b** | | | | | | | | |
| Full Symmetric | 36.8 | 66.9 | 90.0 | 22.7 | 77.7 | 71.3 | 23.6 | 57.5 |
| LightRetriever Dense | 31.9 | 59.8 | 88.7 | 18.5 | 70.3 | 71.5 | 26.7 | 52.7 |
| LightRetriever Sparse | 33.1 | 57.9 | 84.6 | 18.1 | 66.7 | 61.1 | 31.7 | 48.0 |
| LightRetriever Hybrid | 33.8 | 62.5 | 88.8 | 19.8 | 71.6 | 76.7 | 32.4 | 54.5 |
| **Qwen2.5-1.5b** | | | | | | | | |
| Full Symmetric | 36.4 | 57.1 | 89.0 | 20.4 | 72.7 | 73.0 | 28.4 | 54.0 |
| LightRetriever Dense | 29.8 | 50.9 | 86.7 | 18.1 | 66.5 | 72.6 | 29.1 | 48.9 |
| LightRetriever Sparse | 32.9 | 51.6 | 81.3 | 17.4 | 66.4 | 64.8 | 31.1 | 47.3 |
| LightRetriever Hybrid | 34.0 | 55.8 | 87.0 | 19.3 | 69.2 | 76.8 | 32.3 | 52.1 |
| **Qwen2.5-3b** | | | | | | | | |
| Full Symmetric | 35.9 | 61.4 | 89.5 | 22.3 | 72.9 | 65.7 | 26.0 | 54.9 |
| LightRetriever Dense | 30.6 | 54.5 | 87.5 | 18.5 | 64.8 | 71.2 | 26.1 | 50.1 |
| LightRetriever Sparse | 33.2 | 54.2 | 82.2 | 17.6 | 67.6 | 57.0 | 26.4 | 47.5 |
| LightRetriever Hybrid | 34.2 | 58.5 | 87.8 | 19.5 | 69.1 | 74.2 | 29.8 | 52.8 |
| **Qwen2.5-7b** | | | | | | | | |
| Full Symmetric | 37.2 | 63.7 | 89.7 | 23.7 | 77.2 | 69.1 | 25.3 | 56.6 |
| LightRetriever Dense | 30.7 | 55.9 | 87.6 | 19.5 | 67.9 | 64.1 | 24.1 | 50.6 |
| LightRetriever Sparse | 33.9 | 55.5 | 78.7 | 18.3 | 69.0 | 60.5 | 28.5 | 48.2 |
| LightRetriever Hybrid | 34.4 | 60.1 | 87.9 | 21.1 | 71.1 | 73.1 | 29.4 | 53.8 |

Table 16: Detailed main results (nDCG@10) on CMTEB Retrieval dev sets.

| Task / Dev, nDCG@10 | Cmedqa | Covid | DuReader | Ecom | MMarco | Medical | T2 | Video | Avg (# 8) |
|---|---|---|---|---|---|---|---|---|---|
| Static Embedding | 7.5 | 45.7 | 37.9 | 33.9 | 34.9 | 15.7 | 35.8 | 38.9 | 31.3 |
| BM25 | 13.7 | 86.6 | 57.1 | 45.1 | 48.3 | 32.1 | 60.5 | 62.7 | 50.8 |
| BGE-m3-dense | 31.0 | 77.2 | 82.9 | 57.5 | 76.9 | 51.5 | 80.8 | 54.4 | 64.0 |
| BGE-m3-sparse | 24.5 | 76.0 | 71.4 | 50.3 | 59.2 | 44.0 | 71.7 | 58.5 | 57.0 |
| LLM2Vec-llama3-8b | 35.2 | 16.5 | 80.7 | 54.4 | 76.5 | 54.6 | 65.2 | 52.4 | 54.4 |
| E5-Mistral | 34.2 | 73.1 | 87.0 | 46.0 | 74.8 | 52.8 | 80.7 | 45.4 | 61.8 |
| **Llama3.2-1b** | | | | | | | | | |
| Full Symmetric | 32.3 | 74.2 | 84.6 | 53.8 | 73.5 | 50.5 | 78.0 | 63.1 | 63.8 |
| LightRetriever Dense | 27.2 | 67.6 | 80.8 | 51.0 | 67.9 | 43.0 | 73.4 | 58.5 | 58.7 |
| LightRetriever Sparse | 17.3 | 64.3 | 74.4 | 51.0 | 63.3 | 39.4 | 69.3 | 52.2 | 53.9 |
| LightRetriever Hybrid | 27.2 | 69.7 | 81.8 | 55.5 | 69.8 | 44.1 | 75.9 | 62.3 | 60.8 |
| **Llama3.2-3b** | | | | | | | | | |
| Full Symmetric | 36.8 | 75.6 | 87.9 | 57.7 | 75.7 | 53.8 | 81.1 | 60.5 | 66.1 |
| LightRetriever Dense | 29.8 | 70.2 | 84.0 | 51.7 | 69.2 | 46.5 | 75.7 | 56.4 | 60.4 |
| LightRetriever Sparse | 17.2 | 65.4 | 76.1 | 47.4 | 65.0 | 42.5 | 68.5 | 42.0 | 53.0 |
| LightRetriever Hybrid | 29.9 | 71.5 | 84.8 | 54.5 | 71.2 | 47.2 | 77.1 | 57.7 | 61.7 |
| **Llama3.1-8b** | | | | | | | | | |
| Full Symmetric | 39.5 | 77.1 | 89.5 | 58.9 | 78.1 | 57.6 | 84.1 | 56.2 | 67.6 |
| LightRetriever Dense | 32.1 | 71.0 | 85.7 | 53.0 | 70.2 | 49.0 | 78.5 | 52.0 | 61.4 |
| LightRetriever Sparse | 15.6 | 70.4 | 82.9 | 47.3 | 63.9 | 42.2 | 75.1 | 41.3 | 54.8 |
| LightRetriever Hybrid | 32.1 | 72.8 | 86.6 | 55.3 | 71.6 | 49.7 | 80.1 | 55.8 | 63.0 |
| **Mistral0.3-7b** | | | | | | | | | |
| Full Symmetric | 38.5 | 74.1 | 88.6 | 54.7 | 75.6 | 56.6 | 82.3 | 56.5 | 65.9 |
| LightRetriever Dense | 30.4 | 64.4 | 83.6 | 47.0 | 64.3 | 46.3 | 74.7 | 48.5 | 57.4 |
| LightRetriever Sparse | 11.5 | 63.2 | 80.5 | 43.4 | 61.7 | 36.7 | 71.3 | 37.4 | 50.7 |
| LightRetriever Hybrid | 30.3 | 67.2 | 84.5 | 50.7 | 67.6 | 46.9 | 76.4 | 51.4 | 59.4 |
| **Qwen2.5-1.5b** | | | | | | | | | |
| Full Symmetric | 37.9 | 78.2 | 87.6 | 56.9 | 75.9 | 54.7 | 78.8 | 57.0 | 65.9 |
| LightRetriever Dense | 31.3 | 73.8 | 85.3 | 52.8 | 71.8 | 49.5 | 75.2 | 55.7 | 61.9 |
| LightRetriever Sparse | 14.5 | 66.7 | 78.1 | 49.3 | 61.6 | 39.1 | 63.5 | 47.0 | 52.5 |
| LightRetriever Hybrid | 31.2 | 74.4 | 86.0 | 56.5 | 73.4 | 50.5 | 76.4 | 61.6 | 63.8 |
| **Qwen2.5-3b** | | | | | | | | | |
| Full Symmetric | 39.4 | 80.1 | 89.5 | 61.9 | 77.8 | 58.7 | 84.4 | 63.6 | 69.4 |
| LightRetriever Dense | 32.7 | 76.0 | 86.8 | 56.7 | 73.2 | 51.6 | 80.9 | 59.5 | 64.7 |
| LightRetriever Sparse | 12.8 | 70.4 | 80.2 | 48.8 | 61.1 | 39.4 | 72.4 | 47.9 | 54.1 |
| LightRetriever Hybrid | 32.6 | 77.0 | 87.6 | 58.3 | 74.0 | 51.6 | 82.0 | 62.8 | 65.7 |
| **Qwen2.5-7b** | | | | | | | | | |
| Full Symmetric | 41.3 | 81.4 | 91.0 | 61.1 | 80.0 | 59.8 | 85.5 | 60.3 | 70.1 |
| LightRetriever Dense | 34.3 | 74.6 | 88.0 | 55.7 | 75.4 | 53.2 | 81.6 | 57.2 | 65.0 |
| LightRetriever Sparse | 12.5 | 72.5 | 82.4 | 48.3 | 65.2 | 40.4 | 73.3 | 45.2 | 55.0 |
| LightRetriever Hybrid | 34.0 | 76.7 | 88.6 | 58.0 | 76.2 | 53.9 | 83.0 | 61.4 | 66.5 |

A.10 ADDITIONAL RESULTS ON COIR AND BRIGHT

Benchmarks of our query-lightweight design on CoIR (Code Retrieval) and BRIGHT (Reasoning-intensive Retrieval) have been conducted using three backbones (Llama-3.2-1B, Llama-3.1-8B, Qwen-2.5-7B). Results are shown below in Table 17 and 18.

As code retrieval and reasoning-intensive retrieval are more challenging out-of-domain (OOD) tasks requiring deep query or structure understanding, our proposed query-lightweight encoders are less performant compared to strong LLM embedding models like E5-Mistral (-6.2pp on CoIR, -6.3pp on BRIGHT), which is expected in our work. Our method is more suitable for normal relevance-based retrieval. Queries requiring complicated instructions or code understanding are not suitable to be applied to the query-lightweight encoder.

Table 17: Retrieval performance (nDCG@10) on the CoIR benchmark.

| Model | Apps | CosQA | Synth-etic-Text2-SQL | Code-Search-Net | Code-SNCC | Code-Trans-Contest | Code-Trans-DL | Stack-Over-Flow-QA | Code-Feed-Back-MT | Code-Feed-Back-ST | Avg |
|---|---|---|---|---|---|---|---|---|---|---|---|
| BM25 | 1.0 | 14.0 | 16.9 | 26.8 | 34.7 | 50.1 | 8.7 | 56.8 | 34.7 | 54.3 | 29.8 |
| BGE-M3 | 7.4 | 22.7 | 48.8 | 43.2 | 47.6 | 47.9 | 31.2 | 61.0 | 33.5 | 49.9 | 39.3 |
| E5-Mistral | 21.3 | 31.3 | 66.0 | 54.3 | 65.3 | 82.6 | 33.2 | 91.5 | 33.7 | 72.7 | 55.2 |
| **LightRetriever** | | | | | | | | | | | |
| Llama3.2-1B | 11.2 | 30.1 | 52.0 | 60.1 | 58.6 | 57.7 | 27.9 | 77.8 | 29.6 | 64.5 | 47.0 |
| Llama3.1-8B | 18.1 | 29.5 | 54.5 | 57.8 | 59.0 | 59.7 | 25.7 | 82.0 | 33.8 | 68.1 | 48.8 |
| Qwen2.5-7B | 18.4 | 30.0 | 54.0 | 57.7 | 61.8 | 61.5 | 22.3 | 84.4 | 33.1 | 66.3 | 49.0 |

Table 18: Retrieval performance (nDCG@10) on the BRIGHT benchmark.

| Model | Bio. | Earth. | Econ. | Psy. | Rob. | Stack. | Sus. | Leet. | Pony | AoPS | TheT. | TheQ. | Avg |
|---|---|---|---|---|---|---|---|---|---|---|---|---|---|
| BM25 | 18.9 | 27.2 | 14.9 | 12.5 | 13.6 | 18.4 | 15.0 | 24.4 | 7.9 | 6.2 | 10.4 | 4.9 | 14.5 |
| BGE-large | 11.7 | 24.6 | 16.6 | 17.5 | 11.7 | 10.8 | 13.3 | 26.7 | 5.7 | 6.0 | 13.0 | 6.9 | 13.7 |
| E5-Mistral | 18.6 | 26.0 | 15.5 | 15.8 | 16.3 | 11.2 | 18.1 | 28.7 | 4.9 | 7.1 | 26.1 | 26.8 | 17.9 |
| **LightRetriever** | | | | | | | | | | | | | |
| Llama3.2-1B | 13.7 | 21.4 | 11.2 | 11.1 | 18.5 | 13.0 | 12.7 | 15.2 | 4.0 | 2.9 | 6.5 | 6.0 | 11.4 |
| Llama3.1-8B | 14.2 | 22.3 | 12.6 | 11.1 | 17.1 | 13.9 | 15.0 | 16.6 | 8.1 | 0.7 | 4.0 | 5.7 | 11.8 |
| Qwen2.5-7B | 19.3 | 24.0 | 13.8 | 13.1 | 14.1 | 11.3 | 14.3 | 11.8 | 5.2 | 1.1 | 5.8 | 5.7 | 11.6 |

