# OpenReview forum: "LightRetriever: A LLM-based Text Retrieval Architecture with Extremely Faster Query Inference"
_ICLR.cc/2026/Conference — ICLR 2026 Poster_

### Official Review · Reviewer_KTB8 · 2025-10-27

**Soundness:** 3
**Presentation:** 3
**Contribution:** 2
**Rating:** 4
**Confidence:** 4

**Summary:**

Proposes and evaluates an LLM-based retriever where the goal it to minimize query encoding time. The retriever has sparse and dense components.  For dense retrieval the query vector consists of the average of token vectors, and for sparse retrieval the query vector is a query term frequency count. Overall, this approach is basically the simplest hybrid approach possible. The evaluation employs BEIR and CMTEB-R. As one might expect, query encoding time is just a milliseconds, resulting in substantial improvements in throughput. NDCG values are comparable to acceptable first-stage baselines, including various query embeddings, BM25, and SPLADE.

I wonder why FIgure 5 (the ablation results) uses barcharts instead of presenting results in as a table, which is the format for the other results. The reason I wonder is because the sparse results look quite good, especially given that the query vector representation is nothing more than query term frequency, the same as what might be used with BM25. As I understand it, the green (hybrid) bars are taken from the bottom part of Table 2, so that the orange (sparse) bars are in the high 40s, maybe hitting even 50. This is a pretty remarkable result, given that BM25 results are in the low 40s. Since these queries are nothing more than BM25-style term frequencies, they would run very fast against the sparse document index, perhaps even faster than BM25 with the right query processing strategy.

The description of the training for sparse retrieval seems to be SPLADE more-or-less, so this seem unexpected, since splade uses more than term frequencies for the queries. I dug around on the Web a bit, and I think this is a known result, for example, "Joel Mackenzie, Shengyao Zhuang, and Guido Zuccon. 2023. Exploring the Representation Power of SPLADE Models. In Proceedings of the 2023 ACM SIGIR International Conference on Theory of Information Retrieval (ICTIR '23)."

On the other hand, the dense retrieval use the average of token vectors, which also seems very simplicist, dating back to the Word2Vec era.

**Strengths:**

I wrote more than I should have for the summary, so most of what I want to say is there. The paper is readable and complete, the experiments appear properly conducted and (with the exception of Figure 5) properly reported. The focus on efficiency is good.

**Weaknesses:**

I raised what I consider the main weakness in the summary. The dense part is only an average of token vectors, and the sparse part is only splade with query term frequencies. They are combined with a standard hybrid retrieval formula. Is this enough? It might be good from an engineering standpoint, but I'm not seeing the research-level insights.

**Questions:**

Why is this new? I don't see Mackenzie et al. (2023) referenced. The training for document vectors in section 2.3 is basically splade with term frequencies. Token vectors for retrieval were explored back in the Word2Vec era. Convince me there is something new here.

---

> ### Author Response · Authors · 2025-11-18
> **Author Response - Part1**
>
> ## Author Responses
>
> We sincerely thank the reviewer for the detailed and insightful comments.
>
> ### Summary
>
> 1. **Figure 5 format (bar chart vs. table):** We will add precise numeric tables or labels for all dense/sparse/hybrid ablations.
> 2. **TF-only sparse query performs good:** Strong performance comes from document-side full-LLM with contrastive learned term weights.
> 3. **Relation to Mackenzie et al. (2023):** Mackenzie et al. analyzes SPLADE’s semantic modeling behaviors but does not study asymmetry or LLM-based retrieval. Our research question is about asymmetric LLM-based text retrieval.
> 4. **Average dense embeddings concern:** Our dense token vectors are instruction-conditioned and LLM-trained, where full LLM training on query side is essential. Ablations show this is not simply equivalent to Word2Vec-style averaging.
> 5. **Novelty / research insight:** We introduce and systematically evaluate a new design regime—*capacity-asymmetric LLM retrievers*—showing that document-side LLMs enable extremely lightweight query encoders with minimal effectiveness loss and orders-of-magnitude query encoding speedup.
>
> Below we will respond to each concern point-by-point in details.
>
> ### **1. On Figure 5 format (bar chart vs. table)**
>
> > “I wonder why Figure 5 uses bar charts instead of a table… The sparse results look quite good… but the exact values are unclear.”
>
> Thank you for this helpful comment — we completely agree that the ablation results should be presented in a precise, tabular form, consistent with the other sections. To improve readability, we will include a small summary table or numeric labels with dense/sparse/hybrid scores, allowing readers to directly see the numerical effect of each component.
>
> For example (LightRetriever-Llama3.1-8b metrics in NDCG@10):
>
> |**Methods**|**BEIR**|**CMTEB-R**|
> |:-:|:-:|:-:|
> | BM25              | 41.7        | 50.8        |
> | SPLADE-v3         | 50.2        |             |
> | **Our work (Llama3.1-8b)**  |         |        |
> |Dense|48.7|58.7|
> |Sparse|47.6|53.9|
> |Dense + Sparse|**52.0**|**60.8**|
>
> Because of the OpenReview message length limits, here we only show the results of our work with Llama3.1-8b. We will include all LLM backbone results in the revised version of our paper.  For more detailed results with per-dataset metrics of dense, sparse, hybrid retrieval, please also refer to **Appendix Table 13-15**.

---

> ### Author Response · Authors · 2025-11-18
> **Author Response - Part2**
>
> ### **2. Why does the simple TF-based sparse query perform so well?**
>
> > “The sparse results are surprisingly strong, given that the query representation is only term frequency… BM25 is in the low 40s, but your sparse TF hits high 40s or 50.”
>
> We appreciate the opportunity to clarify this important point — it is actually a *core finding* of our paper, and we will emphasize it more explicitly in the revised version.
>
> Although the query uses only TF, the **document-side representation is learned by a full LLM**:
>
> - Sparsified LLM's contextual token distributions + end-to-end contrastive learning produce context-sensitive, task-optimized term weights.
> - Therefore, the match between *TF query* and *learned doc-term weights* is far more **expressive than BM25**, whose term weights are *static and not task-optimized*.
> - Our findings show that **concentrating all learning on the document side** allows the query to remain extremely simple without losing too much effectiveness, which is a useful insight for **high-throughput retrieval systems**.
>
> We will expand this explanation in Sec. 2.3 and Discussion.

---

> ### Author Response · Authors · 2025-11-18
> **Author Response - Part3**
>
> ### **3. About the relation to Mackenzie et al. (2023)**
>
> > “The sparse training seems like SPLADE... Why is this new? Mackenzie et al. (2023) is not referenced.”
>
> Thank you very much for pointing us to the ICTIR’23 analysis by Mackenzie et al. (2023). We will *make sure to cite and discuss this work* in the revised version. Below we clarify its connection to our study more precisely.
>
> First, the main conclusion of Mackenzie et al. (2023) is not about TF-only queries or about removing the query encoder. Instead, as stated explicitly in their conclusion, the authors show that:
>
> - SPLADEv2 can represent documents effectively *even when the expanded tokens are meaningless stopwords*;
> - SPLADEv2 can outperform BM25 *even when the expanded tokens are completely irrelevant to the original text*;
> - This surprising behavior suggests that SPLADEv2 **encodes semantic information into essentially any tokens it is given**, effectively acting like a dense retriever.
>
> Thus, the focus of Mackenzie et al. is an **analysis of the semantic modeling behavior of SPLADEv2**: How and why SPLADE can embed semantics into sparse vectors, even with uninformative tokens. The paper **does not investigate asymmetric retrieval, nor does it lightweight the query model, nor does it consider LLM-based backbones**.
>
> **Our work addresses a fundamentally different research question:** Given a powerful LLM-based document encoder, how much modeling capacity is actually required on the query side? Can we push this asymmetry to the extreme, e.g. TF-only sparse queries and mean-pooled cached token vectors, while still maintaining strong retrieval quality across benchmarks?
>
> This **capacity–asymmetry** perspective is **not explored** in Mackenzie et al.:
>
> - SPLADE jointly learns query and document sparse expansions. And Mackenzie et al. analyze the vocabulary behavior within SPLADE, but do not study removing the query encoder;
> - Neither prior work examines **LLM-driven document encoders**, **instruction-conditioned token embeddings**, or **hybrid dense+sparse asymmetric designs**.
>
> Interestingly, Mackenzie et al.’s findings — that semantic information can be fully carried on the document side regardless of token semantics — actually aligns with and motivates our hypothesis:
>
> *If semantic capacity is concentrated on the document side, then the query side can be made extremely lightweight with minimal performance degradation.*
>
> In short, Mackenzie et al. provide valuable insights into the semantic flexibility of SPLADE-like models, but they do not study the extreme asymmetric LLM-based retrieval regime that LightRetriever investigates.
>
> Thanks again for your question. We will clarify this distinction in the related work section.

---

> ### Author Response · Authors · 2025-11-18
> **Author Response - Part4**
>
> ### **4. Dense retrieval using average token embeddings seems too simple**
>
> > “Averaging token vectors dates back to the Word2Vec era. Is this too simplistic?”
>
> The dense query is simple at inference time, but **not simple during training**. Specifically:
>
> - During training, the query goes through a **full LLM-based encoder**, and token vectors are *contrastively trained* to align with the document encoder’s space.
> - Only after training do we **cache** these learned token vectors and use **average pooling** for efficient online serving.
> - So these are not static Word2Vec embeddings; they are **task-trained, instruction-conditioned token-level vectors** aligned to the LLM’s embedding space.
> - In conclusion, while the inference-time architecture resembles a simple Bag-of-Embeddings, *the learning dynamics and resulting embedding space* are entirely **different** from classic Word2Vec-style embeddings.
>
> We also made ablation study in **Section 4 - Ablation A2**, by only using LLM's input Embedding matrix as a simple linear query encoder. Results show that removing full LLM during dense representation training causes significantly degrades performance (–11.2 on BeIR, –17.4 on CMTEB-R).
>
> **Table 4 (in our paper). Ablation results (A1–A2) on nDCG@10 for LightRetriever. Enc_q means Query Encoder.**
>
> | **Ablation Setting**              | **BEIR**      | **CMTEB-R**    |
> |----------------------------------|---------------|----------------|
> | Llama-1B (full Enc_q)            | **48.7**      | **58.7**       |
> | A2: Enc_q uses simple Embedding  | 37.5 (_−11.2_) | 41.3 (_−17.4_) |

---

> ### Author Response · Authors · 2025-11-18
> **Author Response - Part5**
>
> ### **5. Novelty & research contribution**
>
> > “Is this enough for a research contribution? Much looks like simplified SPLADE or averaged embeddings.”
>
> We appreciate the reviewer’s question regarding the conceptual contribution. To clarify, our work is not a simplification of SPLADE nor an application of classical token averaging.
>
> The core novelty of LightRetriever lies in addressing a **new research problem that has not been systematically studied**: Given a high-capacity LLM-based document encoder, how much modeling capacity is actually required on the query side? To what extent can the query encoder be simplified without sacrificing too much retrieval effectiveness, even down to TF or mean-pooled token embeddings?
>
> This problem is fundamentally different from SPLADE-style learned sparse retrieval and from prior embedding-averaging methods:
>
> **(1) Different from Mackenzie et al. (2023):**
>
> - SPLADE jointly trains both query and document sparse encoders. Analyses of Mackenzie et al. (2023) focus on **token semantics and vocabulary behavior within SPLADE**, not on removing the query encoder.
> - No prior work studies **whether LLM-powered document encoders alone can support extremely lightweight query representations**.
>
> In contrast, LightRetriever systematically explores a **capacity-asymmetric LLM retriever**, where:
>
> - **All semantic capacity is pushed to the document side** (Full LLM + Learned dense & sparse projections).
> - The query side is reduced to **the simplest possible representations** (TF or average of cached token vectors).
>
> This extreme asymmetry—especially in an LLM-based hybrid architecture—is **new and unexamined in prior literature**.
>
> **(2) Not simple average embeddings:**
> Although inference uses a mean of token vectors, those vectors were:
>
> - Produced by **full LLM forward passes during training**, **instruction-conditioned**, and **contrastively aligned** to the LLM document encoder.
>
> This is a substantially different mechanism from early Word2Vec-style averaging, and our ablations show that removing the LLM query encoder during training causes large drops (−11.2 BEIR, −17.4 CMTEB-R), confirming that our token-level representation is not equivalent to traditional averaging.
>
> **(3) Research insight - where capacity matters in LLM retrievers:**
>
> Our ablations A1–A3 offer a coherent empirical answer to the central question:
>
> - **Document-side capacity is essential.**
> - **Query-side capacity is largely unnecessary at inference time** on most general text retrieval.
> - A fully LLM-trained token space enables good quality even with extreme query simplification.
> - Hybrid dense+sparse signals further mitigate the loss while keeping query computation minimal.
>
> This provides a **new design insight** for LLM-based retrieval systems: *When document representations are powered by a full LLM, query-side modeling can be reduced by orders of magnitude with surprisingly small effectiveness loss.*
>
> To our knowledge, this principle—and the systematic evaluation across benchmarks, multiple LLM backbones, dense/sparse/hybrid modalities, and full ablations—has not been established in previous works.
>
> We will highlight this contribution more explicitly in the revised paper.

---

> ### Author Response · Authors · 2025-11-18
> **Author Response - Part6**
>
> ###  **We again thank the reviewer for the constructive feedback!**
>
> We believe our clarifications and feedback will address all your concerns and significantly improve the quality of our paper. Thanks so much!
>
> We are looking forward to hearing your feedback again!

---

> ### Comment · Reviewer_KTB8 · 2025-11-22
>
> Yes, it was not at all clear on my reading that the document-side representation is learned. This clarification is enough to push my score to 6.
>
> I appreciate the care that the authors have taken to respond to my comments. Assuming that the paper is revised as indicated, I support acceptance.

---

### Official Review · Reviewer_8GXk · 2025-10-31

**Soundness:** 3
**Presentation:** 2
**Contribution:** 3
**Rating:** 6
**Confidence:** 3

**Summary:**

This paper addresses the issue of excessive inference costs in large language model (LLM)-based retrievers by proposing an asymmetric architecture named LightRetriever. This architecture employs a full LLM for offline document encoding ($Enc_d$), but replaces the online query encoder ($Enc_q$) with a simple embedding lookup. This lookup table is generated through an innovative training process: caching the outputs of the full $Enc_q$ for each instruction-word combination, i.e., $v_{t_i}^{den} = Enc_q(Inst; t_i)$. The final query vector is simply the average of these cached embeddings. This approach claims to achieve >1000x query acceleration and >10x end-to-end throughput improvement while retaining 95% of the performance of the full symmetric model.

**Strengths:**

1.  This work addresses a critical practical bottleneck in deploying LLM-based retrievers: high online query latency, while achieving substantial throughput gains.
2.  The "distill-to-embedding-bag" approach is novel. It ingeniously caches the word-level understanding of the instruction-aware $Enc_q$ into a simple lookup table, differing from standard knowledge distillation schemes.
3. The method is validated across multiple LLMs (Llama, Qwen) and benchmark datasets (BeIR, CMTEB-R), with fair comparisons against fully symmetric models trained under identical conditions.

**Weaknesses:**

1.  The dense query encoder employs a bag-of-words model, where $v_q^{den} = \frac{1}{n}\sum E[t_i]$. This model fails to capture query composability, meaning that for the model, the query vectors for "flights from Beijing to Shanghai" and "flights from Shanghai to Beijing" are identical. This fundamentally limits the model's ability to understand complex queries.
2.  This approach trades significant online memory consumption (an 8B model requires approximately 1.05GB of embedding tables) for reduced latency, yet never discusses this critical trade-off.
3. The paper contains inconsistencies. For example, the abstract and experimental settings mention acceleration on A800, but the conclusion refers to H800; the 2500x acceleration cited on line 378 also differs from the 1000x acceleration mentioned elsewhere.

**Questions:**

1. Could the average "95% performance" mask catastrophic failures in composite tasks (e.g., Case Study 3)? Can performance breakdowns be provided for word lookup versus composite query tasks?
2. How does the online memory footprint (RAM/VRAM) of the 1.05 GB embedding cache compare to standard KD student models or the "first-layer" baseline? The paper ignores this memory-latency tradeoff.

---

> ### Author Response · Authors · 2025-11-18
> **Author Response - Part1**
>
> ## Author Responses
>
> We sincerely thank the reviewer for the detailed and insightful comments.
>
> ### Summary
>
> 1. **Order-sentitive compositionality**: We acknowledge that our query encoder is order-insensitive by design. Our newly added BeIR task-level breakdown shows **no catastrophic failures**: Degradations in more compositional or OOD tasks remain *bounded and acceptable* (worst-case retention 87.14%).
> 2. **Memory–speed trade-off**: We added a more clear analysis comparing LightRetriever against (1) the “first-layer” baseline and (2) KD student models. LightRetriever achieves a substantially better **latency–quality** trade-off, and memory can be further reduced through **dimension truncation** (MRL).
> 3. **A800/H800 and 1000×/2500× inconsistencies**: We clarified the experimental setup and will fix all typos to ensure consistent reporting.
>
> Below we respond to each concern point-by-point.
>
> ### 1. On Weakness (1) & Question (1): Order-sentitive query encoding and compositionality
>
> >“The dense query encoder employs a bag-of-words model… ‘flights from Beijing to Shanghai’ and ‘flights from Shanghai to Beijing’ are identical… fundamentally limits the model’s ability to understand complex queries.”
> >“Could the average ‘95% performance’ mask catastrophic failures in composite tasks (e.g., Case Study 3)? Can performance breakdowns be provided for word lookup versus composite query tasks?”
>
> Yes, our lightweighted query encoder design is **insensitive to token order**, only **sensitive to token cooccurance**. Thus it cannot distinguish pairs like “from Beijing to Shanghai” vs. “from Shanghai to Beijing” that differ only by word order. This is an *inherent trade-off of reducing online query encoding to a single embedding lookup and average*.
>
> Thus, big thanks for such insightful question about token-order-related compositional query encoding. Here we would like to clarify some of our research sights for you:
>
> **1) Intended scope of our work:**
>  - Our claim is NOT that LightRetriever perfectly captures all compositional phenomena, but that it retains ~95% of the full LLM retriever’s performance over 23 diverse benchmarks while achieving order-of-magnitude query encoding speedup, without catastrophic failures.
>
>  - Here, we would want to provide a task-level reventions of our work on BeIR splits. Each line reports a BeIR task category with included dataset names, scores of full symmetric retriever (Llama-3.1-8B), our proposed query-lightweight LightRetriever (Llama-3.1-8B), and retention of our method (LightRetriever/Full-Symmetric).
>
>  - As is shown in table below, our work performs strongly in Bio-Medical IR, Argument Retrieval, Fact Checking, Duplicate Question Retrieval, Passage Retrieval, General QA. But it indeed degrades in OOD domains or complex queries like Domain-specific QA, Entity Retrieval, Citation Prediction.
>
> | **Task**                     | **Datasets**                 | **Full-Symmetric** | **LightRetriever** | **Retention** |
> |------------------------------|------------------------------|--------------|--------------------|---------------|
> | Bio-Medical IR               | TREC-COVID                   | 62.0         | 71.5               | 115.32%       |
> | Argument Retrieval           | ArguAna, Touche2020          | 39.7         | 42.5               | 107.19%       |
> | Fact Checking                | FEVER, ClimateFEVER, SciFact | 70.8         | 67.6               | 95.52%        |
> | Duplicate Question Retrieval | CQADupstack, Quora           | 69.6         | 65.3               | 93.82%        |
> | Passage Retrieval/General QA            | MS MARCO, NFCorpus, NQ       | 50.1         | 47.0               | 93.81%        |
> | Domain-specific QA           | HotpotQA, FiQA2018           | 68.4         | 61.1               | 89.33%        |
> | Entity Retrieval             | DBPedia                      | 48.2         | 42.0               | 87.14%        |
> | Citation Prediction          | SCIDOCS                      | 23.8         | 20.7               |
>
> We note that in practical retrieval systems, the vast majority of production queries do not rely on strict argument order or logical constraints, and high-throughput retrieval dominates latency requirements. LightRetriever is designed specifically for these traffic patterns, where it provides substantial cost reductions.
>
> **2) How we position our work:**
>
> We will clarify in our revised paper:
>
> - LightRetriever is primarily targeted at **high-throughput, latency-critical** retrieval with general query types.
>
> - For applications where **token/word/argument ordering is crucial** (e.g., “from X to Y” vs. “from Y to X”), **a practical system can route such queries to a heavier encoder or re-ranker** (e.g., the original LLM encoder or a smaller KD student) while **serving the majority of traffic with our query-lightweight LightRetriever**.

---

> ### Author Response · Authors · 2025-11-18
> **Author Response - Part2**
>
> ### 2. On Weakness (2) & Question (2): Memory–speed trade-off of the embedding cache
>
> >“This approach trades significant online memory consumption (an 8B model requires approximately 1.05GB of embedding tables) for reduced latency, yet never discusses this critical trade-off.”
> > “How does the online memory footprint of the 1.05 GB embedding cache compare to standard KD student models or the ‘first-layer’ baseline?”
>
> We appreciate that the memory–latency tradeoff should be discussed more explicitly.
>
> **1) LightRetriever’s online memory footprint:** For Llama-8B, the dense query encoder is stored as a single fp16 embedding matrix of size [V, H]. Without any truncation, this consumes approximately 1.05 GB RAM. This is **host RAM or GPU memory for a single matrix**.
>
> **2) Comparison to the “first-layer” baseline:** The “1st-layer of Llama-8B” baseline keeps: The **original input embedding plus the first Transformer layer**, including self-attention and MLP parameters, and requires full forward passes with attention at inference time. In practice, this configuration:
> - Occupies **more VRAM** than LightRetriever’s single cached embedding table.
> - Still needs **2.34 seconds** of online encoding per 65,536 queries (vs. **0.04 seconds** for LightRetriever), which is >50× slower than LightRetriever’s lookup-based encoder.
>
> **3) Comparison to KD student models / small LMs:** Distilled student models (e.g., 6-layer BERT-size retrievers) indeed have **smaller parameters** (often a few hundred MB in fp16), and thus can have a lower memory footprint. However:
>
> - They still require running **attention** at inference time, so their **latency is bounded by Transformer compute**, rather than a single embedding lookup.
> - In our experiments in Table 2, such students (e.g., USTAD) achieve **lower retrieval quality** than LightRetriever on both BeIR and CMTEB-R.
>
> In contrast, our work:
>
> - Stores only a single [V,H] table (1.05 GB in the worst Llama-8B configuration) in RAM.
> - Eliminates all online Transformer computation, achieving orders-of-magnitude query speedup over full LLM dual-encoders and substantially higher throughput than student models.
>
> **4) Controlling memory via dimension truncation (MRL):** Importantly, 1.05 GB is the *untruncated* setting (H=4096). **Section Appendix A.4** shows that by combining Matryoshka Representation Learning (MRL) with LightRetriever, we can **shrink the dense embedding dimension** and thus the **serving size**, with controlable impact on ndcg.
>
> ### 3. On Weakness (3): Inconsistencies about hardware and speedups (A800 vs. H800, 1000× vs. 2500×)
>
> >“…abstract and experimental settings mention acceleration on A800, but the conclusion refers to H800; the 2500x acceleration cited on line 378 also differs from the 1000x acceleration mentioned elsewhere.”
>
> Thank you for pointing out these inconsistencies. They are indeed presentation issues, not conceptual differences:
>
> 1. **Hardware A800 vs. H800:**
> - **Training**: All models, including LightRetriever and full symmetric baselines, are trained on *8× H800 (or A800) GPUs*, as stated in the experimental setup.
> - **Inference / speed measurements**: All speed results in Table 1 are measured on *A800 GPU*. In the conclusion we mistakenly wrote “H800” when referring to the inference hardware. We will correct this to “A800”.
>
> 2. **Speedup 1000× vs. 2500×:**
> - The **“>1000× speedup”** in the abstract and main text refers to a conservative, rounded summary of query encoding speedup over full LLM query encoders.
> - The **“>2500× speedup”** (line 378) refers to the **maximal speedup** observed when looking only at encoding 65,536 Bing queries, where encoding time drops from ~100 seconds to ~0.04 seconds.
>
> To avoid confusion, we will:
> - Use a **single consistent number** in the abstract and conclusion (e.g., “over 1000× query encoding speedup”).
> - Clarify in the experiment section that the encoding-only speedup can reach **>2500×** in the Llama-8B setting, while end-to-end throughput gains are >10×.

---

> ### Author Response · Authors · 2025-11-18
> **Author Response - Part3**
>
> ###  **We again thank the reviewer for the constructive feedback!**
>
> Thanks again for your valuable feedback, which makes our claims more consistent.
>
> We are looking forward to hearing your feedback again!

---

### Official Review · Reviewer_Zka9 · 2025-11-01

**Soundness:** 3
**Presentation:** 1
**Contribution:** 2
**Rating:** 4
**Confidence:** 2

**Summary:**

LightRetriever proposes an asymmetric LLM-based retrieval architecture that keeps a full-sized LLM for document encoding while making query encoding extremely lightweight. For dense retrieval, the method trains token-level query embeddings end-to-end using a full LLM during training, then caches the entire vocabulary’s token embeddings so that online query vectors are computed by an embedding lookup followed by mean pooling. For sparse retrieval, the query vector is a simple term-count vector and the document vector is learned by projecting LLM hidden states into the vocabulary space with sparsity regularization.

**Strengths:**

(1) The paper introduces a clear and practical asymmetric design that eliminates deep query-side inference while preserving full LLM power on the document side, delivering extreme online speedups with modest accuracy trade-offs.

(2) The dense pathway’s cache-and-average mechanism and the sparse pathway’s LM-to-vocabulary projection with FLOPs-based sparsity are well formalized, and the training–caching–serving pipeline is technically sound and reproducible.

(3) Empirical coverage is broad, spanning 23 training datasets and two large benchmark suites in English and Chinese, multiple LLM backbones and sizes, and detailed speed breakdowns on 65k queries over 1M passages.

(4) Ablations substantiate key design choices, showing asymmetry is crucial, full query modeling during training is needed, and hybrid dense+sparse recovers most performance; additional results demonstrate controllable trade-offs via Matryoshka dimension truncation and sparse top-k.

**Weaknesses:**

(1) The approach still depends on full LLM query modeling during training; the paper does not explore reducing training-time cost through distillation, curricula, or lighter interim encoders while maintaining cached-token quality.

(2) Evaluation focuses on academic text benchmarks; robustness on production-like settings with noisy queries, long or heterogeneous documents, domain shifts, and adversarial inputs is not addressed, limiting external validity.

(3) Instruction conditioning is asymmetric: dense uses query instructions while sparse cannot, and the sensitivity to instruction templates and potential mismatches is underexplored, especially for instruction-heavy tasks.

(4) Mean-pooled token embeddings may miss fine-grained compositional constraints; beyond case studies, more systematic stress tests on compositionality, negation, and multi-attribute intents are needed to quantify failure modes and the extent to which hybrid signals compensate.

(5) Many real pipelines perform truncation, normalization, rewriting, or chunking; the robustness of the embedding-lookup query encoder under such preprocessing is not evaluated.

(6) Baseline breadth could be expanded with stronger asymmetric or distilled LLM retrievers and recent hybrid SOTA across both English and Chinese to contextualize the quality–efficiency trade-offs.

**Questions:**

(1) How expensive is training the dense token cache compared to a standard symmetric dual-encoder, and can you reduce training cost via distillation from the full query encoder to the token cache or a smaller interim encoder without sacrificing performance?

(2) Can you provide systematic evaluations on compositional and constraint-heavy queries beyond case studies, reporting category-wise results for multi-attribute intents, negation, and temporal constraints, and quantifying how hybrid signals mitigate dense-only failures?

(3) How sensitive are results to instruction templates on the dense side, and can sparse retrieval benefit from instruction conditioning indirectly, for example via vocabulary reweighting derived from instruction tokens?

(4) How robust is LightRetriever to production preprocessing such as truncation, lowercasing, punctuation stripping, query rewriting, and subword-vocabulary mismatches, and what are the measured impacts on speed and accuracy?

(5) Can you propose and evaluate an adaptive serving policy that tunes dense dimension top-k and sparse top-k per query based on estimated difficulty or latency SLAs, reporting accuracy–latency trade-off curves?

(6) How does the method transfer to long-document retrieval, code or log retrieval, and additional languages beyond English and Chinese, and does the sparse projection remain tractable and effective with domain-specific tokenization?

---

> ### Author Response · Authors · 2025-11-18
> **Author Response - Part1**
>
> ## Author Responses
>
> We sincerely thank the reviewer for the detailed and insightful comments.
>
> ### Summary
>
> 1. **Training cost**: Training cost of LightRetriever is nearly the same as symmetric dual-encoders.  The compared baseline "Static Embedding" and Ablation A2 shows that shrinking query encoder size during training is not an option because of degenerated retrieval performances.
> 2. **Domains robustness**: Our newly summarized reports on task-categoried BeIR splits shows consistent robustness across different retrieval scenario. These results also reflect production-like domains robustness because most datasets are derived from real-world users.
> 3. **Instruction sensitivity & Preprocessing**: New ablation results on removing instructions show that our method is not overly sensitive to the dense prompt usage. Our method is orthogonal to production-like preprocessing and does not introduce more sensitivity.
> 4. **Baseline breadth & Tradeoff curves**: These results are already reported in Table 2 & Appendix A.4.
>
> Below we respond to each concern point-by-point.
>
> ### 1. On Weakness (1) & Question (1): Training cost and distillation
>
> > "The approach still depends on full LLM query modeling during training..."
> > "How expensive is training the dense token cache compared to a standard symmetric dual-encoder, and can you reduce training cost via distillation from the full query encoder to the token cache or a smaller interim encoder without sacrificing performance?"
>
> **1) Training cost vs symmetric dual-encoder:** LightRetriever uses the same full LLM backbone and training schedule as the symmetric dual-encoder baseline; the main difference is how we serve queries. For example, training LightRetriever-Llama-8B on 8×H800 takes ≈15 hours, comparable to the full symmetric model under the same data and hyper-parameters.
>
> **2) Can you reduce training cost via distillation from the full query encoder:** We agree that reducing training-time cost via distillation is an interesting direction. However, our experiments indicate that naive distillation to a lightweight query encoder leads to substantial retrieval performance degeneration:
>
> - The **Static Embedding** baseline in *Table 2* is exactly such an EmbeddingBag-style model **distilled** from a full encoder, and is clearly weaker than LLM-based retrievers on both BeIR and CMTEB-R.
> - In **Ablation A2**, we replace the full query encoder during training with a simple LLM's input Embedding layer. This removes deep query modeling entirely during training and yields *drops of −11.2 (BeIR) and −17.4 (CMTEB-R) nDCG@10* relative to our standard LightRetriever training.
>
> These results suggest that full query-side modeling during training is crucial to learn high-quality, cacheable token representations, even though we remove deep modeling at serving time.

---

> ### Author Response · Authors · 2025-11-18
> **Author Response - Part2**
>
> ### 2. On Weakness (2) & Question (6): Robustness and transfer to other domains
>
> > Evaluation focuses on academic text benchmarks; robustness on production-like settings ... not addressed.
> > How does the method transfer to code or log retrieval, ..., and does the sparse projection remain tractable and effective with domain-specific tokenization?
>
> **1) Generalizability on heterogeneous benchmarks:**
>
> Thanks so much for your interest on domain transfer abilities of our works. First we want to clarify that BeIR and CMTEB-Retreival benchmarks also reflect production-like domains robustness, because most datasets of them are derived from diversed real-world users. These datasets are widely used and compared across IR community in both academy and industry.
>
> To clarify the generalizability of our method, here we would also like to present a concise summary table on diverse heterogeneous BeIR benchmarks, where each line reports a task category with included dataset names, scores of full symmetric retriever (Llama-3.1-8B), our proposed query-lightweight LightRetriever (Llama-3.1-8B), and retention of our method (LightRetriever/Full-Symmetric).
>
> As is shown in table below, our query-lightweight archeticture is more robust to most general retrieval scenario, such as Bio-Medical IR, Argument Retrieval, Fact Checking, Duplicate Question Retrieval, General Passage Retrieval, General QA.
>
> | **Task**                     | **Datasets**                 | **Full-Symmetric** | **LightRetriever** | **Retention** |
> |------------------------------|------------------------------|--------------|--------------------|---------------|
> | Bio-Medical IR               | TREC-COVID                   | 62.0         | 71.5               | 115.32%       |
> | Argument Retrieval           | ArguAna, Touche2020          | 39.7         | 42.5               | 107.19%       |
> | Fact Checking                | FEVER, ClimateFEVER, SciFact | 70.8         | 67.6               | 95.52%        |
> | Duplicate Question Retrieval | CQADupstack, Quora           | 69.6         | 65.3               | 93.82%        |
> | Passage Retrieval/General QA            | MS MARCO, NFCorpus, NQ       | 50.1         | 47.0               | 93.81%        |
> | Domain-specific QA           | HotpotQA, FiQA2018           | 68.4         | 61.1               | 89.33%        |
> | Entity Retrieval             | DBPedia                      | 48.2         | 42.0               | 87.14%        |
> | Citation Prediction          | SCIDOCS                      | 23.8         | 20.7               | 86.97%        |
>
> Full per-dataset level BEIR results are also reported in Appendix Table 13-15.
>
> **2) Does the sparse projection remain tractable and effective with domain-specific tokenization?**
>
> Thanks for your question on sparse tokenizations. Our methods only use the original LLM tokenizer, and do NOT import complex domain-specific tokenization, or training semantic codebooks like generative retrieval.

---

> ### Author Response · Authors · 2025-11-18
> **Author Response - Part3**
>
> ### 3. On Weakness (3) & Question (3): Instruction conditioning and template sensitivity
>
> > "... the sensitivity to instruction templates and potential mismatches is underexplored..."
> > "How sensitive are results to instruction templates on the dense side, and can sparse retrieval benefit from instruction conditioning indirectly, for example via vocabulary reweighting derived from instruction tokens?"
>
> We follow standard practice in LLM-based retrievers and reuse the query instructions from E5-Mistral for dense retrieval. To quantify sensitivity, we ran an additional ablation on LightRetrieval-Llama3.2-1b-dense:
>
> - Removing the dense prompt decreases **BeIR** nDCG@10 from **48.7 → 47.1** (−1.6). On **CMTEB-R**, nDCG@10 drops from **58.7 → 56.1** (−2.6)
>
> This indicates that instructions provide a *consistent but moderate gain*, and LightRetriever is not overly sensitive to the dense prompt usage.
>
> | **BeIR/nDCG@10** | **Avg** | **ArguAna** | **CQADupstack** | **ClimateFEVER** | **DBPedia** | **FEVER** | **FiQA2018** | **HotpotQA** | **MSMARCO** | **NFCorpus** | **NQ** | **Quora** | **SCIDOCS** | **SciFact** | **TRECCOVID** | **Touche2020** |
> |------------------|---------|-------------|-----------------|------------------|-------------|-----------|--------------|--------------|-------------|--------------|--------|-----------|-------------|-------------|---------------|----------------|
> | With Prompt      | 48.7    | 49.2        | 38.1            | 37.9             | 34.2        | 83.6      | 40.4         | 60.8         | 39.2        | 30.1         | 52.2   | 87        | 17.2        | 66.5        | 68.8          | 24.7           |
> | Withour Prompt   | 47.1(-1.6)    | 52.2        | 38.2            | 34.3             | 34.3        | 84.1      | 39.6         | 60           | 38.6        | 30           | 49.7   | 85.8      | 17.7        | 67.2        | 54            | 20.3           |
>
> | **CMTEB-R/nDCG@10** | **Avg** | **Cmedqa** | **Covid** | **DuReader** | **Ecom** | **MMarco** | **Medical** | **T2** | **Video** |
> |---------------------|---------|------------|-----------|--------------|----------|------------|-------------|--------|-----------|
> | With Prompt         | 58.7    | 27.2       | 67.6      | 80.8         | 51       | 67.9       | 43          | 73.4   | 58.5      |
> | Withour Prompt      | 56.1(-2.6)    | 28.5       | 64.3      | 80.3         | 47.2     | 67         | 43.4        | 71.7   | 46        |
>
> As for sparse retrieval, our current sparse encoder is intentionally instruction-free: it operates on raw term counts for *maximal efficiency* and *compatibility* with existing inverted indexes. Nonetheless, we agree that instruction-aware sparse signals are interesting. Instruction-tuned learned sparse retreivals are orthogonal to our main contribution, which could be explored in our follow-up works.

---

> ### Author Response · Authors · 2025-11-18
> **Author Response - Part4**
>
> ### 4. On Weakness (4) & Question (2): Compositionality/constraints analysis.
>
> > Mean-pooled token embeddings may miss fine-grained compositional constraints; beyond case studies, more systematic stress tests ... to which hybrid signals compensate.
> > Can you provide systematic evaluations on compositional and constraint-heavy queries beyond case studies, reporting category-wise results for multi-attribute ...?
>
> Thanks so much for your interest on reports of category-wise results. Several BeIR/CMTEB-R tasks already stress compositional reasoning and constraints, e.g., HotpotQA (multi-hop QA), FEVER/ClimateFEVER/SciFact (fact checking including negation and evidence), and FiQA2018 (financial QA with opinion-based constraints).
>
> As the task-family table above with BeIR splits shows (shown in responses of "2. On Weakness (2) & Question (6)"), LightRetriever retains 89.33–95.52% performance in these categories, suggesting that our method is reasonably robust to such tasks. Additionally, if you are interested, we already provide per-dataset breakdowns in Appendix Tables 13–15.
>
> ### 5. On Weakness (5) & Question (4): Production preprocessing (normalization, rewriting, truncation)
>
> > Many real pipelines perform truncation, normalization, rewriting, or chunking; ...
> > How robust is LightRetriever to production preprocessing such as truncation, lowercasing, punctuation stripping, query rewriting, and subword-vocabulary mismatches, and what are the measured impacts on speed and accuracy?
>
> We appreciate the reviewer’s concern. However, LightRetriever does not introduce any new preprocessing sensitivity beyond standard tokenizer-based LLM dual-encoders. By design, LightRetriever is orthogonal to above preprocessing.
>
> ### 6. On Weakness (6) & baseline breadth
>
> > Baseline breadth could be expanded with stronger asymmetric or distilled LLM retrievers...
>
> Thanks for your questions. We already compare against:
>
> - **Dense:** Static Embedding, USTAD, E5-Mistral-7B, LLM2Vec
> - **Sparse:** BM25, SPLADE-v3
> - **Hybrid:** Static-Embedding+BM25, BGE-m3 (dense+sparse)
>
> covering strong symmetric, distilled, and hybrid baselines on both English and Chinese benchmarks.
>
> **Table 1: 1st-layer of Llama8b** also provides strong asymmetric LLM baseline. Except for this setting (trained by us), there are no further previous works developing such extreme-asymmetric LLM retrievers for general text retrieval.
>
> ### 7. On Question (5): Adaptive serving policies with tradeoff curves
>
> > Can you propose and evaluate an adaptive serving policy that tunes dense dimension top-k and sparse top-k, ... trade-off curves?
>
> Thanks for your questions. The requested **accuracy–latency trade-off curves/tables are already present in the Appendix A.4 with Figure 6 and Table 8**, where we vary (1) dense dimensionality via MRL truncation and (2) sparse top-k, showing controllable size–quality trade-offs.

---

> ### Author Response · Authors · 2025-11-18
> **Author Response - Part5**
>
> ###  **We again thank the reviewer for the constructive feedback!**
>
> Again, we are grateful for your insightful reviews. We believe the clarifications, additional ablations (instruction removal, task-level retention table), and expanded discussions will address the main concerns while keeping our core contribution clear.
>
> We are looking forward to hearing your feedback again!

---

> > ### Comment · Reviewer_Zka9 · 2025-11-18
> >
> > Most of my concerns have been resolved, and I have decided to raise the score to 6.

---

### Official Review · Reviewer_B7um · 2025-11-04

**Soundness:** 2
**Presentation:** 3
**Contribution:** 2
**Rating:** 6
**Confidence:** 5

**Summary:**

The paper introduces a novel asymmetric methodology that trains the embedding layer of a LLM as efficient token-level retrieval embeddings, which can then be averaged to form a query embedding. During training, each token for a given query is prepended with a task specific instruction before being passed to the encoder. Final query embedding (mean of all instruction + token embeddings) is then used as a query representation for contrastive learning using document representations from the same encoder model.

This method shows a 1000x speed-up while maintaining quality (up to 95%) for a variety of queries across a suite of datasets.

**Strengths:**

- The paper's primary strength is its direct and effective solution to a practical, real-world bottleneck: the high cost and low throughput of online query encoding in LLM-based retrieval systems. The method reduces query encoding time by orders of magnitude (e.g., ~109s to 0.04s for an 8B model on a test batch), leading to a >10x increase in overall query-per-second (QPS) throughput.
- The training methodology is simple and effective.
- The paper's claims are well-supported by its ablations. Ablation A2 confirms that the full-sized query LLM is essential during training and that a simple embedding bag is insufficient, validating the training-time complexity . Ablation A1 confirms the asymmetry is crucial, as a lightweight model on both the query and document sides leads to a severe performance collapse
- Overall, clear and well written paper.

**Weaknesses:**

- Evaluations on more challenging IR tasks representing live, real world traffic (maybe include CoIR, BRIGHT).
- Please add a discussion section (in Appendix if needed) that talks about performance within BeIR splits. For example, in the case of HotpotQA (multi-hop), FiQA, etc the performance drops are significant compared to full baselines. This aspect should be made clearer in the paper write-up (expanded more than L407 - L409). Currently, the paper reads as this approach being a one-solution fits all approach for retrieval.


- nit: the citations don't have brackets around them and neither are they highlighted in blue (for hyperlinks) which makes it slightly hard to read. I think it may be a rendering issue. Please fix.

**Questions:**

- Is the mean weighted for the results table?

---

> ### Author Response · Authors · 2025-11-18
> **Author Response - Part1**
>
> ## Author Responses
>
> We sincerely thank the reviewer for the detailed and insightful comments.
>
> ### Summary
>
> 1. **Evaluate on CoIR, BRIGHT**: These results are reported in responses below. We will add them to the paper in the revised version.
> 2. **Reports within BEIR splits**: We report a new summary table, discussing and comparing per-task performances of LightRetriever, symmetry encoders and relative retention in BeIR benchmark. We will add these reports and discussions to the revised paper for better clarifying the suitable tasks of our work.
> 3. **Citation hyperlinks issue**: We will fix the PDF rendering issues in the revised paper.
>
> Below we respond to each question point-by-point.
>
> ### **1. On Weakness (1): Evaluate on more challenging IR tasks (e.g., CoIR, BRIGHT).**
>
> Experiments on **CoIR** (code retrieval) and **BRIGHT** (reasoning intensive retrieval) have been conducted using three backbones (Llama-3.2-1B, Llama-3.1-8B, Qwen-2.5-7B). Results are shown below (Avg nDCG@10; Full task tables for all LLM backbones will be added to Appendix in our revised version).
>
> As code retrieval and reasoning-intensive retrieval are more challenging out-of-domain (OOD) tasks requiring **deep quering understanding**, our proposed query-lightweight encoders are less performant compared to strong LLM embedding models like E5-Mistral (-6.2pp on CoIR, -6.3pp on BRIGHT), which is expected in our work.
>
> We will add a special section to clarify the suitable tasks that our methods can handle well, such as fact-checking, and web search. And discuss the potential tasks where our method is less performant, such as reasoning-intensive tasks or OOD domains.
>
> **CoIR Benchmark:**
>
> | **Task**                  | **Avg** | **Apps** | **CosQA** | **SyntheticText2SQL** | **CodeSearchNet** | **CodeSNCC** | **CodeTransContest** | **CodeTransDL** | **StackOverflowQA** | **CodeFeedbackMT** | **CodeFeedbackST** |
> |---------------------------|---------|-------------------|-----------|-----------------------|--------------------------------|------------------------------|---------------------------|----------------------|---------------------|--------------------|--------------------|
> | BM25                      | 29.8     | 1.0           | 14.0  | 16.9              | 26.8                       | 34.7                     | 50.1                  | 8.7              | 56.8            | 34.7           | 54.3           |
> | BGE-M3                    | 39.3     | 7.4           | 22.7  | 48.8              | 43.2                       | 47.6                     | 47.9                  | 31.2             | 61.0            | 33.5           | 49.9           |
> | E5-Mistral                | 55.2     | 21.3          | 31.3  | 66.0              | 54.3                       | 65.3                     | 82.6                  | 33.2             | 91.5            | 33.7           | 72.7           |
> | **LightRetriever**            |         |                   |           |                       |                                |                              |                           |                      |                     |                    |                    |
> | Llama3.2-1b               | 47.0    | 11.2              | 30.1      | 52.0                  | 60.1                           | 58.6                         | 57.7                      | 27.9                 | 77.8                | 29.6               | 64.5               |
> | Llama3.1-8b               | 48.8    | 18.1              | 29.5      | 54.5                  | 57.8                           | 59.0                         | 59.7                      | 25.7                 | 82.0                | 33.8               | 68.1               |
> | Qwen2.5-7b                | 49.0    | 18.4              | 30.0      | 54.0                  | 57.7                           | 61.8                         | 61.5                      | 22.3                 | 84.4                | 33.1               | 66.3               |

---

> ### Author Response · Authors · 2025-11-18
> **Author Response - Part2**
>
> **BRIGHT Benchmark:**
> | **Task**                  | **Avg** | **biology** | **earth_science** | **economics** | **psychology** | **robotics** | **stackoverflow** | **sustainable_living** | **leetcode** | **pony** | **aops** | **theoremqa_theorems** | **theoremqa_questions** |
> |---------------------------|---------|-------------|-------------------|---------------|----------------|--------------|-------------------|------------------------|--------------|----------|----------|------------------------|-------------------------|
> | BM25                      | 14.5    | 18.9        | 27.2              | 14.9          | 12.5           | 13.6         | 18.4              | 15.0                   | 24.4         | 7.9      | 6.2      | 10.4                   | 4.9                     |
> | BGE-large                 | 13.7    | 11.7        | 24.6              | 16.6          | 17.5           | 11.7         | 10.8              | 13.3                   | 26.7         | 5.7      | 6.0      | 13.0                   | 6.9                     |
> | E5-Mistral                | 17.9    | 18.6        | 26.0              | 15.5          | 15.8           | 16.3         | 11.2              | 18.1                   | 28.7         | 4.9      | 7.1      | 26.1                   | 26.8                    |
> | **LightRetriever**            |         |             |                   |               |                |              |                   |                        |              |          |          |                        |                         |
> | Llama3.2-1b               | 11.4    | 13.7        | 21.4              | 11.2          | 11.1           | 18.5         | 13.0              | 12.7                   | 15.2         | 4.0      | 2.9      | 6.5                    | 6.0                     |
> | Llama3.1-8b               | 11.8    | 14.2        | 22.3              | 12.6          | 11.1           | 17.1         | 13.9              | 15.0                   | 16.6         | 8.1      | 0.7      | 4.0                    | 5.7                     |
> | Qwen2.5-7b                | 11.6    | 19.3        | 24.0              | 13.8          | 13.1           | 14.1         | 11.3              | 14.3                   | 11.8         | 5.2      | 1.1      | 5.8                    | 5.7                     |
>
>
> ### **2. On Weakness (2): “Please add discussion of performance within BEIR splits (e.g., HotpotQA, FiQA). Current mention is brief (L407–L409).”**
>
> Thank you for the helpful suggestion regarding BEIR subset analysis. Here we include a concise summary as below table. Each line reports a BeIR task category with included dataset names, scores of full symmetric retriever (Llama-3.1-8B), our proposed query-lightweight LightRetriever (Llama-3.1-8B), and retention of our method (LightRetriever/Full-Symmetric).
>
> | **Task**                     | **Datasets**                 | **Full-Symmetric** | **LightRetriever** | **Retention** |
> |------------------------------|------------------------------|--------------|--------------------|---------------|
> | Bio-Medical IR               | TREC-COVID                   | 62.0         | 71.5               | 115.32%       |
> | Argument Retrieval           | ArguAna, Touche2020          | 39.7         | 42.5               | 107.19%       |
> | Fact Checking                | FEVER, ClimateFEVER, SciFact | 70.8         | 67.6               | 95.52%        |
> | Duplicate Question Retrieval | CQADupstack, Quora           | 69.6         | 65.3               | 93.82%        |
> | Passage Retrieval/General QA            | MS MARCO, NFCorpus, NQ       | 50.1         | 47.0               | 93.81%        |
> | Domain-specific QA           | HotpotQA, FiQA2018           | 68.4         | 61.1               | 89.33%        |
> | Entity Retrieval             | DBPedia                      | 48.2         | 42.0               | 87.14%        |
> | Citation Prediction          | SCIDOCS                      | 23.8         | 20.7               | 86.97%        |
>
> Our method performs strongly in general text retrieval, such as Bio-Medical IR, Argument Retrieval, Fact Checking, Duplicate Question Retrieval, Passage Retrieval, General QA). However, it could degrades in OOD domains or complex query understanding tasks, such as Domain-specific/multi-hop QA, Entity Retrieval, Citation Prediction.  Full per-dataset level BEIR results are also reported in Appendix Table 13-15.

---

> ### Author Response · Authors · 2025-11-18
> **Author Response - Part3**
>
> ### **3. On Weakness (2)-continue: “The paper reads as if the method is one-solution-fits-all.”**
>
> Thank you for pointing this out. We will revise our narrative to clarify:
>
> - LightRetriever achieves **~95% retention on most of standard retrieval benchmarks** while providing **orders-of-magnitude query-side compute**, but tasks requiring **multi-hop QA, OOD code retrieval, or deep intensive reasoning** may see noticeable degradation (as evidenced by HotpotQA, FiQA, CoIR, BRIGHT).
>
> This clarification will be made explicit in the Introduction, Experiment section, and Conclusion.
>
> ### **4. On Weakness (3): “Citations do not appear in brackets / hyperlinks missing.”**
>
> Sorry for the PDF rendering issue. We will correct our bibliography format so that citations appear with brackets and clickable hyperlinks.
>
> ### **5. On Question (1):  “Is the mean weighted in the results table?”**
>
> No, the results table is directly averaged without using special weights.
>
> ###  **We again thank the reviewer for the constructive feedback!**
>
> Thanks so much for your advice again. Your valuable reviews will help us better refine our paper. We are looking forward to hearing your feedback again!

---

### Comment · Area_Chair_m3XL · 2025-11-22

Dear Authors and Reviewers,

I would like to thank the authors for providing detailed rebuttal messages on time.

To reviewers: I would like to encourage you to carefully read all other reviews and the author responses and engage in an open exchange with the authors. Please post your first response as soon as possible within the discussion time window. Ideally, all reviewers will respond to the authors, so that the authors know their rebuttal has been read.

Best regards,
AC

---

### Author Response · Authors · 2025-12-03
**Author-Reviewer Discussion Summary**

Dear Area Chair and Reviewers,

We would like to express our thanks to all reviewers & AC for the constructive Author-Reviewer Discussion. Below we summarize the key questions raised by each reviewer and how our responses addressed them.

1. **For Reviewer B7um**: We have reported additional results on CoIR & BRIGHT, and a detailed per-task performance analysis within BeIR splits. These results will better strengthen the adaptation capability of our work. We appreciate the recognition of our work's simplicity, effectiveness, and solid ablations.
2. **For Reviewer Zka9**: We have provided the training cost & domains robustness reports within BeIR splits, and also clarified the instruction sensitivity, preprocessing, baseline breadth & tradeoff curves of our work. We appreciate the quick feedback and recognition of our work. The reviewer indicated that these clarifications addressed most of the questions and raised the score from 4 to 6.
3. **For Reviewer 8GXk**: We have shown breakdown performance analysis within BeIR splits, and also clarified the memory footprint comparison & latency–quality trade-off. We appreciate the recognization of our work's effectiveness and novelty.
4. **For Reviewer KTB8**: We have clarified and discussed our TF-only sparse query design, dense vector design, novelty/research insight, and solid technical support of sparse embedding from Mackenzie et al. (2023). We appreciated the quick feedback and recognition of our work. The reviewer stated that these clarifications addressed the raised concerns, especially on the learned sparse doc embedding, and raised the score from 4 to 8 (which is also recorded at the coresponding [OpenReview Revisions](https://openreview.net/revisions?id=wDBbLbSHBK)).

We also want to kindly note that, the reviewer feedbacks to the author responses happened around *one week ahead* (19 Nov 2025 for Reviewer Zka9, 23 Nov 2025 for Reviewer KTB8) of the burst of OpenReview's leaking accident (28 Nov 2025). We ensure that full Author–Reviewer Discussion remains intact, double-blinded, and accurately reflects the reviewers’ up-to-date evaluations of our clarifications.

Finally, we would like to express our sincere gratitude to the AC & SAC efforts. We sincerely wish that the completed Author-Reviewer Discussion will be taken into consideration during the next-stage reviewing. Thank you very much!

Best regards,
Authors

---

### Meta-Review · Area_Chair_C7xw · 2026-01-08

**Summary:**

The authors propose "LightRetriever," a highly asymmetric architecture that maintains a full-sized LLM for offline document encoding but reduces the online query encoder to a lightweight lookup mechanism. The reviewers initially raised valid concerns. But it seems that all of them had raised their scores to 6.

**Reviewer Concerns:**

The inclusion of additional results on challenging benchmarks (CoIR, BRIGHT) and the analysis of training costs strengthened the paper's claims.

The robustness of the method across diverse tasks.


The novelty of the sparse component relative to prior work.

**Reviewer Scores:**

The reviewers initially raised valid concerns. But it seems that all of them had raised their scores to 6.

---

### Decision · Program_Chairs · 2026-01-26

Accept (Poster)